# Structural intermediates observed only in intact *Escherichia coli* indicate a mechanism for TonB-dependent transport

**Thushani D Nilaweera†‡, David A Nyenhuis†§, David S Cafiso***

Department of Chemistry and Center for Membrane Biology, University of Virginia, Charlottesville, United States

*For correspondence:
dsc0b@virginia.edu

†These authors contributed equally to this work

Present address: ‡Genetics and Biochemistry Branch, National Institute of Diabetes and Digestive and Kidney Diseases, National Institutes of Health, Bethesda, United States; §Biochemistry and Biophysics Center, National Heart Lung and Blood Institute, National Institutes of Health, Bethesda, United States

Competing interests: The authors declare that no competing interests exist.

**Abstract** Outer membrane TonB-dependent transporters facilitate the uptake of trace nutrients and carbohydrates in Gram-negative bacteria and are essential for pathogenic bacteria and the health of the microbiome. Despite this, their mechanism of transport is still unknown. Here, pulse electron paramagnetic resonance (EPR) measurements were made in intact cells on the *Escherichia coli* vitamin $B_{12}$ transporter, BtuB. Substrate binding was found to alter the C-terminal region of the core and shift an extracellular substrate binding loop 2 nm toward the periplasm; moreover, this structural transition is regulated by an ionic lock that is broken upon binding of the inner membrane protein TonB. Significantly, this structural transition is not observed when BtuB is reconstituted into phospholipid bilayers. These measurements suggest an alternative to existing models of transport, and they demonstrate the importance of studying outer membrane proteins in their native environment.

## Introduction

The passive permeation of low molecular weight solutes across the outer membrane (OM) of Gram-negative bacteria is typically facilitated by porins. However, many higher molecular weight solutes and trace nutrients, including carbohydrates, iron siderophores, cobalamin, copper, and nickel, are bound and transported across the OM by a family of active transporters that are TonB-dependent (*Bolam and van den Berg, 2018*; *Noinaj et al., 2010*). These TonB-dependent transporters (TBDTs) derive energy from the bacterial inner membrane by interacting with TonB, a transperiplasmic protein that interacts with the inner membrane proteins ExbB and ExbD (*Celia et al., 2016*; *Maki-Yonekura et al., 2018*). This family of transporters has two distinct domains: a β-barrel formed from 22 anti-parallel β-strands, and a core or hatch domain that fills the interior of the barrel. The β-strands on the extracellular surface of the protein barrel are often connected by long loops, while short turns join the strands on the periplasmic interface (see *Figure 1B*). TonB interacts with the transporter through a conserved motif on the N-terminal side of the core termed the Ton box (*Pawelek et al., 2006*; *Shultis et al., 2006*).

The mechanism of transport in TBDTs is presently poorly understood; however, the large size of most substrates and the absence of any obvious pathway for substrate permeation (*Faraldo-Gómez et al., 2003*) have led to proposals that transport is mediated by a significant conformational event that involves a partial rearrangement or full removal of the core domain from the surrounding barrel (*Chimento et al., 2005*). Although many high-resolution structures are available for TBDTs, there is no direct evidence for a major structural change within the core of TBDTs that might indicate a transport mechanism. In the *Escherichia coli* vitamin $B_{12}$ (cobalamin) transporter, BtuB, EPR spectroscopy shows that substrate binding unfolds the Ton box at the N-terminus and extends it into the periplasm, an allosteric event that may facilitate the binding of TonB to BtuB (*Kim et al., 2007*; *Xu et al., 2006*); however, no other significant structural changes have been observed in the core.

**eLife digest** Bacteria must obtain nutrients from their surrounding environment in order to survive. In Gram-negative bacteria, proteins in the outer membrane surrounding the cell actively transport carbohydrates and trace nutrients like iron into the cell's interior. Although the structures of many of these transport proteins have been determined, the mechanism they use to move molecules across the membrane is poorly understood.

To better understand this process, Nilaweera, Nyenhuis and Cafiso examined the structure of BtuB, a transport protein found in the outer membrane of *Escherichia coli* that is responsible for absorbing vitamin $B_{12}$. Previous experiments analyzing the structure of BtuB, and other similar transporters, have been carried out on purified proteins that were extracted from the outer membrane. However, these isolated proteins fail to replicate the transport activity observed in bacterial cells. Nilaweera, Nyenhuis and Cafiso therefore wanted to see how the structure of BtuB changes when it is still enclosed in the membrane of *E. coli*.

This revealed that BtuB undergoes large structural changes when it binds to vitamin $B_{12}$, suggesting that this is an important part of the transport process. However, when purified BtuB was placed into an artificial membrane, these structural changes did not occur. This indicates that the cellular environment in the bacteria is needed for BtuB to carry out its transport role, and explains why previous experiments using purified proteins struggled to see this structural shift.

This work highlights the importance of studying bacterial membrane proteins in their native cell environment. BtuB and similar transporters represent a large family of proteins unique to Gram-negative bacteria that have an impact on human health. Since these proteins are structurally alike, the results of this study may help resolve the transport mechanisms of other proteins, ultimately leading to new ways to control bacterial growth.

High-resolution crystal structures have been obtained for a C-terminal fragment of TonB in complex with BtuB, the ferrichrome transporter FhuA, and the ferrioxamine B transporter FoxA (*Pawelek et al., 2006*; *Shultis et al., 2006*; *Josts et al., 2019*). When TonB binds, the Ton box extends from the core and interacts with the β-sheets of TonB in an edge-to-edge manner. Except for the Ton box, the remainder of the core remains folded and is essentially unchanged. Because TonB binding does not alter the core of BtuB in the BtuB-TonB structure, it has been proposed that TonB alters the core by exerting a mechanical force on the transporter, and current models for transport favor a mechanism where TonB acts by pulling the Ton box thereby unfolding the core (*Gumbart et al., 2007*; *Hickman et al., 2017*; *Sverzhinsky et al., 2015*). Models involving a rotation of TonB have also been proposed (*Klebba, 2016*); however, in FhuA there are four to five unstructured residues between the Ton box and core when TonB is bound, making the transfer of torque from TonB to the core unlikely (*Sarver et al., 2018*). Pulling models have been explored using steered molecular dynamics (MD) (*Gumbart et al., 2007*) as well as single-molecule AFM (atomic force microscopy) pulling experiments (*Hickman et al., 2017*), and these studies indicate that an N-terminal region of the core (up to residue 73) is preferentially unfolded to permit the movement of vitamin $B_{12}$ into the periplasm. This work concludes that the C-terminal region of the core is static and does not unfold during transport, a result that is consistent with denaturation experiments on BtuB (*Flores Jiménez and Cafiso, 2012*).

An important caveat to almost all the structural work on BtuB is that it has been carried out on purified or partially purified protein where the native OM environment is no longer present. Since transport in this family of transporters has never been reconstituted, it has never been established that the isolated, purified, and membrane reconstituted BtuB is capable of transport. Recently, we developed an approach to attach spin labels to either extracellular or periplasmic sites on BtuB in intact cells, thereby permitting EPR measurements to be made under conditions where the protein is known to be functional (*Joseph et al., 2019*; *Nilaweera et al., 2019*). Preliminary measurements made on BtuB indicate that it behaves differently in the intact cell than it does in a purified reconstituted phospholipid system. For example, a substrate-dependent change in the core domain of BtuB involving substrate binding loop 3 (SB3) is observed in situ but is not seen in a detergent-treated OM preparation (*Nilaweera et al., 2019*). Moreover, the extracellular loops of BtuB are also highly

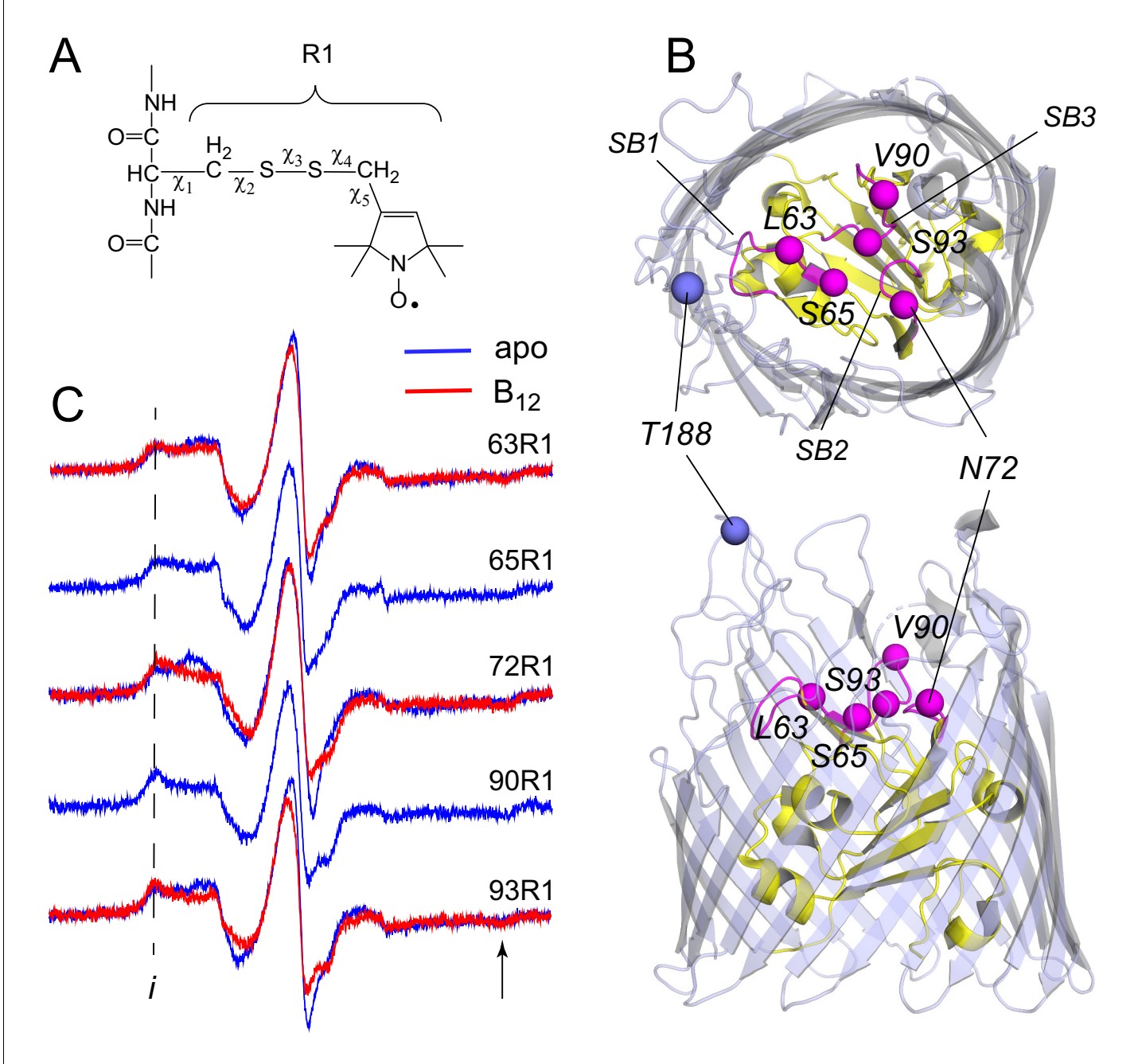

**Figure 1.** Electron paramagnetic resonance (EPR) spectra obtained in vivo from spin labeled core sites on extracellular face of the BtuB. The spin labeled side chain (A) R1 was attached to sites on the extracellular core of BtuB shown in (B). BtuB is shown in both extracellular (top) and side views (PDB ID: 1NQH), with the core in yellow and barrel in light blue. The Cα atom on site 188 on the second extracellular loop is shown, which is used as a reference point for measurements to the core. Labeled Cα sites in the core are shown in magenta, along with substrate binding loops SB1, SB2, and SB3. In (C) EPR spectra are shown in the absence (apo state in blue) and presence of substrate (vitamin $B_{12}$ bound state in red). No change in the spectrum is observed at sites 65 and 90 upon the addition of substrate. The spectra are characterized by well-defined hyperfine extrema ((*i*) and location of arrow), where the difference between these points is approximately 69 Gauss. This value is close to $2A_{zz}$, twice the value of the hyperfine tensor, and indicates that the label is immobilized on the ns time scale. Spectra are the sum of 10–100 Gauss scans.

constrained in the intact cell, and substrate-induced structural changes and structural heterogeneity that is observed for BtuB in proteoliposomes are not observed for BtuB in situ (*Nyenhuis et al., 2020a*).

In the present work, we perform double electron-electron resonance (DEER) on BtuB in intact cells to determine whether structural changes take place in the core domain that are associated with substrate binding and transport. Upon substrate binding, a movement of the core is observed involving sites 90 and 93 in SB3. No other movements in the core are detected. When the ionic lock is broken between site R14 on the C-terminal side of the Ton box and site D316 in the barrel, long-distance components appear upon substrate binding, indicating that SB3 can assume a state where it has moved into the BtuB barrel toward the periplasmic side of the protein. Under these same conditions, other sites on the N-terminal side of the core remain static. Since this ionic interaction would normally be broken upon TonB binding, this structural transition is likely to take place during transport. This result suggests that transport may involve allosteric changes in the C-terminal side of the core upon TonB binding. Remarkably, these substrate-induced structural changes are not observed for purified, membrane reconstituted BtuB, which may be due to the absence of lipopolysaccharide (LPS) or other periplasmic components in the reconstituted system. The importance of the native OM environment provides an explanation for why this structural transition has not been previously observed.

## Results

### Multiple sites in the core region of BtuB may be spin labeled in intact *E. coli*

To investigate movements that might occur in the BtuB core region in situ, pairs of spin labels were placed into the extracellular region of BtuB, with one label located at an outer loop site that is known to be relatively fixed in the cell environment and a second label located at a site in the BtuB core. Previous work has demonstrated that several sites on the extracellular surface of BtuB may be spin labeled in vivo using site-directed cysteines and a standard methanethiosulfonate reagent to produce the side chain R1 (*Figure 1A*). These included multiple sites on the extracellular loops of BtuB (*Nilaweera et al., 2019*; *Nyenhuis et al., 2020a*; *Joseph et al., 2016*) as well as two sites in the core region (*Nilaweera et al., 2019*).

Recent work has also shown that the efficient incorporation of pairs of spin labels to make distance measurements using DEER required the use of a strain deficient in the disulfide bond formation (Dsb) chaperone system (*Nilaweera et al., 2019*). We tested several additional single cysteine mutants in BtuB using a DsbA⁻ strain to determine whether spin labeling of additional sites in the core was possible. Shown in *Figure 1c* are spectra from site 90, which was previously labeled, as wells as four additional sites in the core region. Sites 63 and 65 lie in substrate binding loop 1 (SB1), site 72 lies in substrate binding loop 2 (SB2), and sites 90 and 93 are positioned in SB3. These spectra arise from label having more than one motional component but are dominated by a broad feature that is characteristic of a population of label with hindered motion on the ns time scale, consistent with the confined environment in the extracellular region of the core. For sites 63, 72, and 93, the addition of vitamin $B_{12}$ alters the spectra and increases the population of the immobile component indicating that incorporation of the label at these sites has not prevented the binding of substrate. No significant changes with substrate are seen for sites 65 and 90. At site 90, substrate does bind (see below) and the lack of a change in the EPR spectrum may reflect the fact that in the apo state the label is already highly immobile. Site 65 is also highly immobile, but we cannot exclude the possibly that incorporation of R1 at this site has blocked the binding of vitamin $B_{12}$.

### The apex of the SB3 loop in the BtuB core undergoes a substrate-dependent conformational change

For distance measurements using pulse EPR, each set of spin pairs included a label at position 188 on the 3/4 extracellular loop (the second loop connecting β-strands 3 and 4). This site was chosen as a reference point because previous work in whole cells demonstrated that this loop assumed a well-defined position and exhibited minimal or no movement upon substrate addition (*Nyenhuis et al., 2020a*).

All data were analyzed using LongDistances. Positions of both components were held constant throughout and were set to the average of an initial round of fits where distance was varied. Width

and area for the two components were allowed to vary freely. Error ranges were taken from the output of the fitting routine.

Shown in *Figure 2* are the results for measurements on the V90R1-T188R1 spin pair in cells. Preliminary results from this pair were presented in a previous study demonstrating the use of disulfide chaperone mutants to achieve double labeling of BtuB in whole cells (*Nilaweera et al., 2019*). The background corrected DEER data and resulting distance distributions are shown in *Figure 2b,c*. Both the apo (blue) and vitamin B$_{12}$ bound (red) distributions yield two main intramolecular peaks at

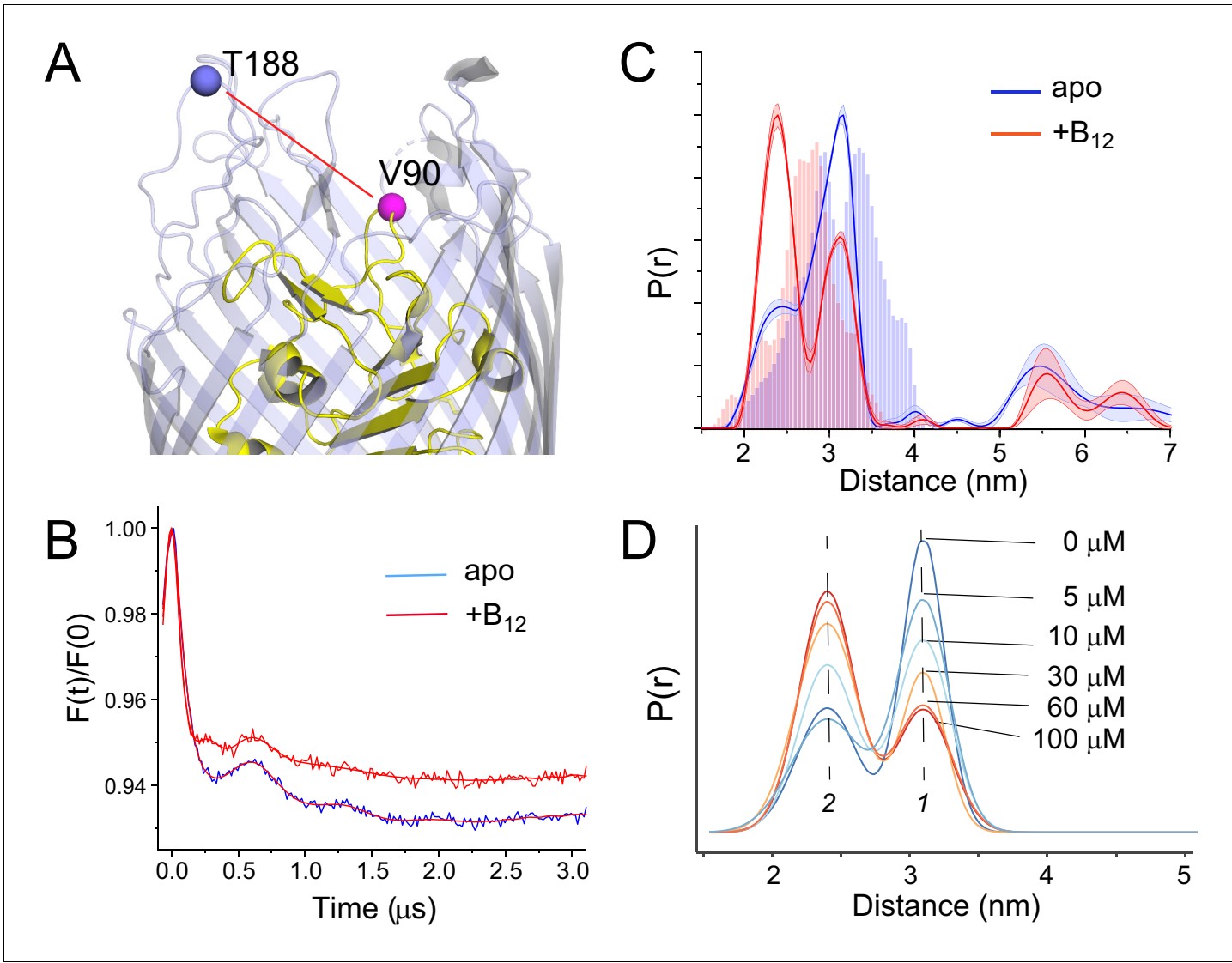

**Figure 2.** Substrate-dependent shifts are detected at the apex of substrate binding loop 3 (SB3) in whole cells. A side view of BtuB (**A**) with the locations of site 90 in SB3 of the core domain (yellow) and site 188 near the apex of the 3/4 extracellular loop (light blue). (**B**) Background corrected double electron-electron resonance (DEER) data for the apo (blue) and substrate bound states (red) of the V90R1-T188R1 spin pair, where the red traces represent the fits to the data. The resulting distance distributions are shown in (**C**) where the histograms represent predicted distance ranges obtained from the *in surfo* crystal structures 1NQG (blue) and 1NQH (red) using the software package MMM (*Jeschke, 2018*). (**D**) This structural change titrates between two states (labeled 1 and 2) with the addition of vitamin B$_{12}$. The conversion between states saturates at concentrations above 60 μM. For the distributions shown in (**C**), data were analyzed using LongDistances v932 using the model-free fitting mode, whereas for distributions in (**D**), data were fit to a two-Gaussian model where position, width, and amplitude were variables in the fit. Errors in the fit to the distributions in (**D**) are shown in *Supplementary file 1*.

The online version of this article includes the following figure supplement(s) for figure 2:

**Figure supplement 1.** Raw and background corrected double electron-electron resonance (DEER) data for V90R1-T188R1.

2.4 and 3.2 nm, with a substrate-dependent shift observed toward the shorter component. Predicted distance distributions were generated from the apo and vitamin $B_{12}$ bound *in surfo* crystal structures of BtuB (PDB IDs: 1NQG and 1NQH) using the program MMM (*Jeschke, 2018*). The distributions generated from these structures also show a shift toward a shorter distance in the substrate bound state, which is due to the unfolding of a helical turn in the SB3 loop (*Chimento et al., 2003*). However, the magnitude of the predicted shift is smaller than that observed by DEER. Because the position of site 188 in the 3/4 extracellular loop of BtuB is not altered with substrate addition in situ (*Nyenhuis et al., 2020a*), this structural change must involve a movement of the SB3 apex or a change in rotamers assumed by R1 at position 90.

We also titrated this structural change by measuring the distance distribution with increasing concentrations of the vitamin $B_{12}$ substrate, where the result is shown in *Figure 2D*. For this analysis data were processed using a model-based approach with two-Gaussian components where the position, width, and amplitude were varied. As seen in *Figure 2D*, there is a strong progressive response to increases in substrate until saturation is reached in the range of 30–60 µM vitamin $B_{12}$. This titration likely reflects substrate loading. Because substrate concentrations greatly exceed the affinity of vitamin $B_{12}$ to BtuB (*Bradbeer et al., 1986*), the saturation point likely reflects the concentration of BtuB in our sample and is roughly consistent with the spin concentrations expected from the EPR signal intensity.

## Substrate-dependent shifts in the extracellular face of the hatch domain are localized to the SB3 loop

To determine whether the structural change observed in the apex of SB3 is limited to this site or part of a broader conformational change across the core domain, we tested additional spin pairs on the extracellular face of the protein using the core sites shown in *Figure 1B*. The spin pairs examined are shown in *Figure 3A,B* and include site 93, which also lies in SB3, as well as sites 63 and 65 in SB1 and site 72 in SB2. The distance distributions that result from these spin pairs are shown in *Figure 3C*, along with the V90R1-T188R1 spin pair. It should be noted that the distance distributions in *Figure 3*, as well as subsequent figures, have been truncated at 5 nm. As seen in the raw data (*Figure 2—figure supplement 1* and *Figure 3—figure supplement 1*), a longer distance component is apparent in some of these dipolar evolutions. This is due to the presence of BtuB-BtuB interactions in the OM that leads to a 6.5 nm distance component (*Nyenhuis et al., 2020b*).

As seen in *Figure 3C*, distance distributions for sites located outside SB3 show little evidence for any structural change upon the addition of substrate, with spin pairs involving sites 63, 65, and 72 having nearly identical distributions for both apo and vitamin $B_{12}$ bound conditions. This lack of a shift in these spin pairs is consistent with work showing that site 188 does not show a substrate-dependent shift in its position in situ (*Nyenhuis et al., 2020a*). In the SB3 loop, however, site 93 at the edge of the loop shows a substrate-dependent change in position along with site 90 at the loop apex, although the change is much larger for site 90 (about 8 Å) than for site 93 (about 4 Å) and in an opposite direction. The structural changes measured by DEER are qualitatively consistent with the predictions from the crystal structures, which show a loss in helical structure and change in position of the loop. For the sites examined here, substrate-dependent changes in the core appear to be confined to the region of the SB3 loop. Interestingly, as we demonstrate below, this substrate-dependent change does not occur when the protein is removed from its native environment.

## Breaking an ionic lock between the BtuB core and barrel triggers a large substrate-dependent change in SB3

As indicated above, BtuB is expected to bind to both substrate and TonB during transport, which will break the ionic lock between R14 in the core and D316 in the barrel (*Shultis et al., 2006*). However, under the conditions of our experiment, BtuB is in large excess relative to TonB, perhaps by a factor of 10–20 or more. As a result, only a small portion of the BtuB would be bound to TonB at any time during our distance measurement.

To determine whether there might be a connection between the structure of the core and this internal ionic lock, we examined the effect of disrupting the R14-D316 interaction on the core by introducing the R14A mutation into the existing pairs of labels between site 188 in the 3/4 extracellular loop and the core. Distance measurements for the apo and vitamin $B_{12}$ bound states made in

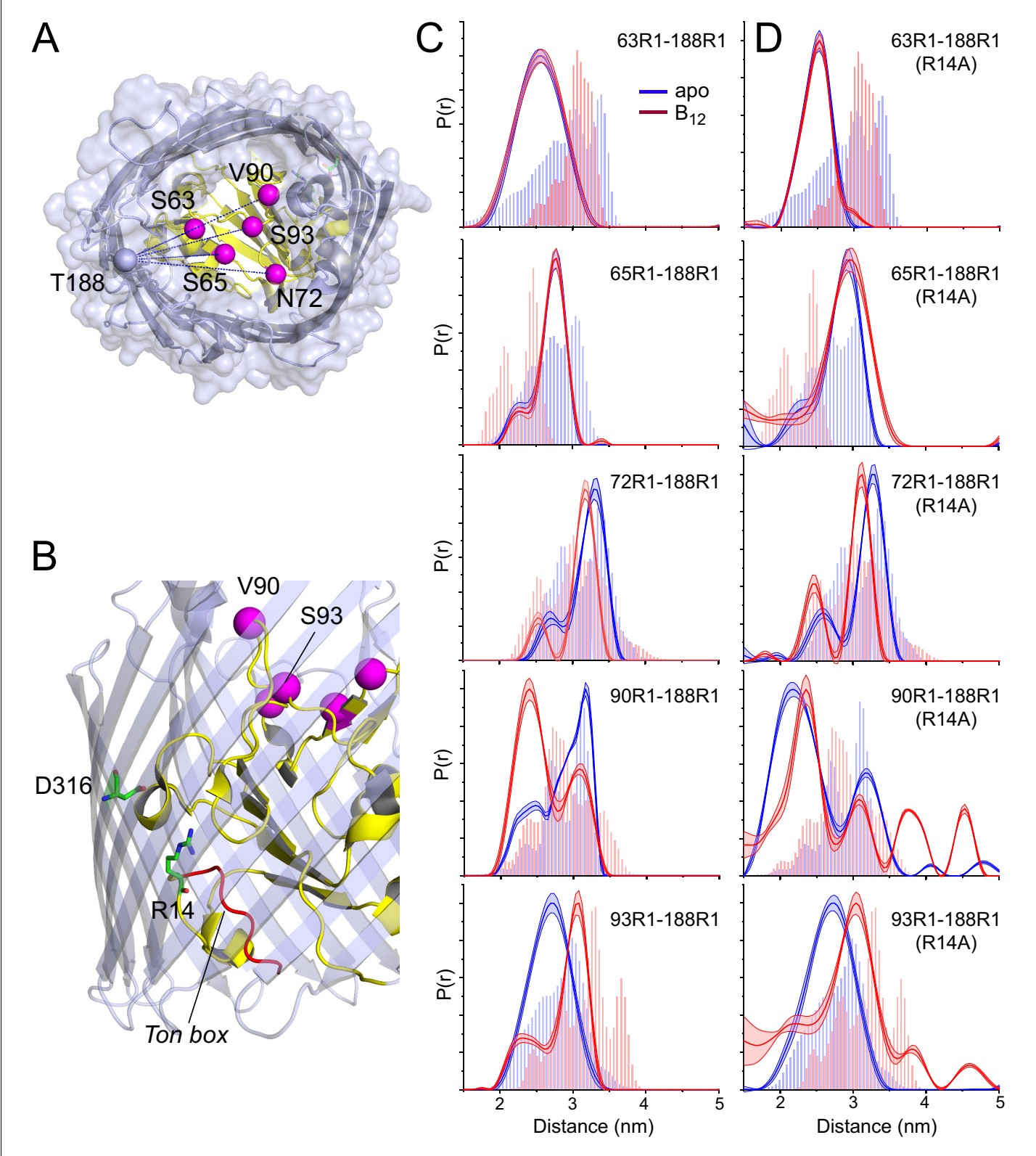

**Figure 3.** Substrate-dependent conformational shifts are limited to substrate binding loop 3 (SB3) and are altered by mutation of the R14-D316 ionic lock. (A) Top view of BtuB (PDB ID: 1NQH) showing the locations of the hatch sites relative to the reference site, 188, in the 3/4 extracellular loop. In (B) the location of the R14-D316 ionic interaction between the core and the barrel is also shown along with the Ton box (red). (C) Distance distributions obtained for hatch: barrel pairs in the apo state (blue) and with substrate (red). (D) Distance distributions obtained for hatch: barrel pairs in the apo

*Figure 3 continued on next page*

*Figure 3 continued*

state (blue) and with substrate (red) in the presence of the R14A mutation. Both V90R1-T188R1 and S93R1-T188R1 spin pairs show additional distances at 3.8 and 4.5 nm in the presence of R14A. Data were analyzed using LongDistances v932 and the model-free fitting regime. The shaded error bands in (C) and (D) represent variation due to background noise, start time, dimensionality, and regularization. Histograms are predicted distances generated from the *in surfo* crystal structures PDB ID: 1NQG (light blue) and PDB ID: 1NQH (light red) using the software package MMM (*Jeschke, 2018*).

The online version of this article includes the following figure supplement(s) for figure 3:

**Figure supplement 1.** Raw and background corrected double electron-electron resonance (DEER) data for loop-core spin pairs.

**Figure supplement 2.** The substrate-dependent movement in substrate binding loop 3 (SB3) promoted by the R14A mutation is observed at site 237.

**Figure supplement 3.** Both wild-type (WT) BtuB and BtuB mutant R14A support growth in minimal media containing vitamin B$_{12}$.

**Figure supplement 4.** Time domain (left), dipolar (middle), and distance (right) data obtained for the V90R1-T188R1 pair with the R14A mutation at various time points show relative stability of these distance components.

the presence of this mutation are shown in *Figure 3D*. For distance distributions involving sites 63, 65, and 72 in SB1 and SB2, the core remains largely unchanged in response to substrate and unchanged by the R14A mutation. However, distance distributions involving sites 90 and 93 in SB3 are altered by breaking the D316/R14 ionic lock.

For distances measured to sites 90 and 93, the R14A mutation has two main effects. First, for the V90R1-T188R1 pair, the substrate-dependent conversion between the 2.4 and 3.2 nm distance components is absent, and the shorter distance now dominates in both apo and vitamin B$_{12}$ bound conditions. Second, for both the V90R1-T188R1 and S93R1-T188R1 pairs, additional distance components are observed in the presence of substrate centered at 3.8 and 4.5 nm. These long components result in a distribution that is substantially broader than that predicted by the crystal structures, and they indicate the formation of a novel conformation of the SB3 loop and an altered substrate binding mode.

The longer distance components that appear for the V90R1-T188R1 and S93R1-T188R1 spin pairs represent a substantial movement of the SB3 loop toward the periplasmic side of BtuB. A movement of SB3 toward the extracellular surface is highly unlikely, in part because movement in this direction would require a major unfolding of the core that we do not observe. For measurements to SB3, there were relatively few positions that were both accessible and within the range of the pulse EPR measurement, but we made measurements to site 90 from site 237, which is located near the apex of the 5/6 extracellular loop. In the apo and vitamin B$_{12}$ bound states, the predicted distances from this site are shorter than 2 nm and are not within a range that can accurately be measured by DEER. The results are shown in *Figure 3—figure supplement 2*. In the absence of the R14A mutation, no clear substrate-dependent shift in the position of SB3 is observed, which is likely due to the short distance involved. However, in the presence of the R14A mutation, a new distance component appears with substrate addition around 3 nm that is beyond the distance range predicted by the crystal structure. This is shorter than the 4.5 nm observed from position 188, which likely reflects differences in the side chain direction and the relative positions of the 3/4 and 5/6 loops.

These ionic lock mutations generate structural states that are not seen in the wild-type protein. However, disrupting the ionic lock does not appear to abolish transport. The BtuB R14A mutant is functional in transport as determined by a growth assay (*Figure 3—figure supplement 3*). This is consistent with earlier work where a mutant containing a dipeptide insertion into one of the barrel strands in BtuB, which should have disrupted the D316-R14A ionic lock, was shown to support transport of vitamin B$_{12}$ (*Lathrop et al., 1995*).

It should be noted that we examined the stability of the substrate-induced conformation of SB3 for times as long as 60 min before freezing and preparing the cells for DEER. The data are shown in *Figure 3—figure supplement 4*, and indicate that the conformations are stable over time, indicating that the label is stable and not being reduced, and that conversion of the transporter back to the apo state does not occur under the conditions of this experiment.

## Substrate-dependent changes in the SB3 loop require a native environment

Two conformations are observed by crystallography for SB3. In the apo structure (PDB ID: 1NQG), SB3 has a single helical turn and shorter conformation that we speculate may be associated with the distance observed by EPR at 3.2 nm for the V90R1-T188R1 spin pair (labeled 1 in *Figure 2D*). In the

vitamin $B_{12}$ bound structure (PDB ID: 1NQH), SB3 is more extended, and this state may be associated with the distance at 2.4 nm (labeled 2 in *Figure 2D*). Computational work suggests that the extended state of SB3 requires the interaction of BtuB with LPS, whereas the shorter helical state of SB3 occurs in the presence of phospholipid (*Balusek and Gumbart, 2016*), suggesting that environment, specifically LPS, may be important in controlling the configuration of SB3.

To test for an environmental effect on SB3, we reconstituted four spin pairs of BtuB into 1-palmi-toyl-2-oleoyl-glycero-3-phosphocholine (POPC) proteoliposomes. These included the V90R1-T188R1 and S93R1-T188R1 spin pairs both in the absence and presence of the R14A mutation. The dipolar evolution data and distance distributions from DEER measurements on these reconstituted BtuB samples are shown in *Figure 4* (the raw time domain data are presented in *Figure 4—figure supplement 1*). In the absence of the R14A mutation, the substrate-dependent conversion to the shorter distance that was seen for the V90R1-T188R1 spin pair in whole cells (*Figure 2C* and *Figure 3C*) is now much more limited, with only a minor shoulder appearing around 2.5 nm. For the S93R1-T188R1 spin pair, the change seen in *Figure 3C* with substrate is largely absent. The 0.4 nm substrate-dependent shift to a longer distance is absent, but the small 2.2 nm distance component is still present.

The behavior of SB3 in the presence of the R14A mutation is also altered in the phospholipid reconstituted system. Rather than increasing the population of the shorter distance component, adding the R14A mutation to the V90R1-T188R1 pair in the reconstituted system results in a single short distance, where the resulting distribution aligns almost perfectly with the predicted distribution from the 1NQH structure. For the S93R1-T188R1 pair, R14A causes a small increase in the short distance component at 2.2 nm. But significantly, for neither spin pair are the longer substrate-induced shifts that were seen in *Figure 3D* observed in the reconstituted system. Thus, when removed from the native OM environment, the structure of SB3 is altered and the large substrate-induced movement of SB3 toward the periplasmic surface in the presence of the R14A mutation is no longer seen.

It should be noted that in our initial work on the V90R1-T188R1 spin pair, we failed to observe a substrate-dependent conformational change in SB3 using an isolated OM preparation where the preparation includes a sarkosyl treatment (*Nilaweera et al., 2019*). This suggests that this detergent treatment of the OM to remove inner membrane components is sufficient to alter the behavior of BtuB. These observations provide an explanation for why these changes in the conformation of SB3 have not been previously observed.

## Mutating either R14, D316, or both alters SB3 conformations and populates a state where SB3 is moved toward the periplasmic surface

In earlier work, we demonstrated that breaking the ionic lock between D316 and R14 altered a conformational equilibrium in the Ton box and promoted its unfolded state (*Lukasik et al., 2007*). In this work, the effect of the R14A mutation on the Ton box equilibrium was comparable to that of a D316A mutation, but slightly enhanced for the dual R14A/D316A mutant.

*Figure 5* shows a result of mutating one or both of R14 and D316 on the substrate-dependent changes in SB3 as measured using the V90R1-T188R1 spin pair (the time domain data are provided in *Figure 5—figure supplement 1*). In the apo state, the distributions for both the R14A and D316A mutants are similar, with peaks falling in the same positions, although the D316A mutant yields more residual area under the 3.2 nm peak than is observed for R14A. In the presence of substrate, both show significant peaks around 3.8 and 4.5 nm, and a significant peak at 2.4 nm in both the apo and vitamin $B_{12}$ bound samples. The double R14A-D316A mutation yields a distance distribution that is more perturbed than either single mutant, with a single broad peak centered around 2.4 nm in the apo state and an increase in the longer distance components in the substrate bound state. Thus, disrupting the D316-R14 ionic lock by mutating one or both residues has a dramatic effect on the conformation of the apex of SB3 and populates a state where this binding loop has moved a significant distance toward the periplasmic interface. The data indicate that this ionic interaction plays a role in mediating allosteric changes within the BtuB core, affecting not only the Ton box equilibrium but the substrate binding loop SB3 on the extracellular surface.

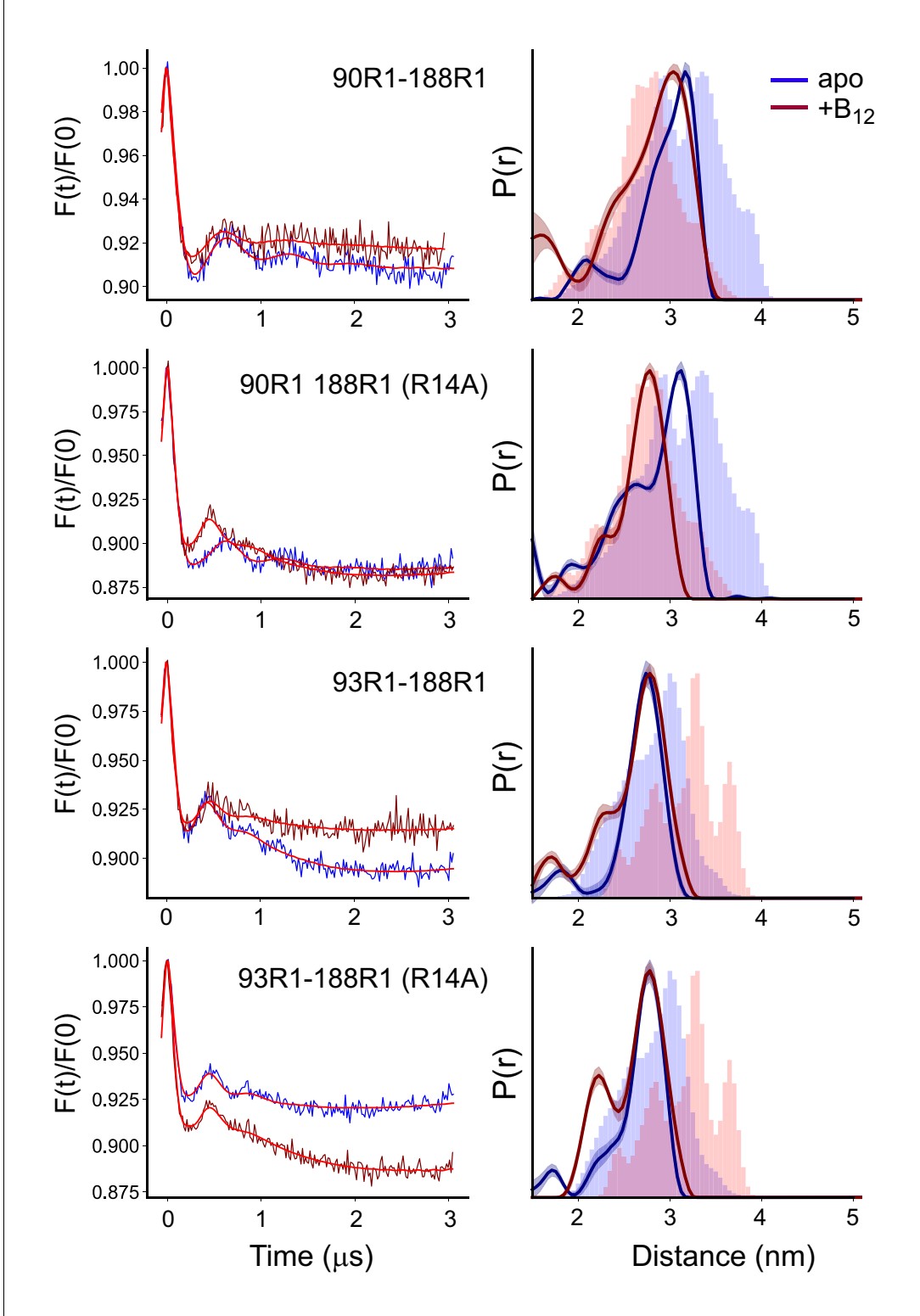

**Figure 4.** The substrate-induced changes in substrate binding loop 3 (SB3) are altered or absent in proteoliposomes. Background corrected double electron-electron resonance (DEER) signals (left) and distance distributions (right) for the V90R1-T188R1 and S93R1-T188R1 pairs involving SB3 in the absence and presence of the R14A mutation, where the labeled BtuB have been reconstituted into 1-palmitoyl-2-oleoyl-glycero-3-phosphocholine (POPC) vesicles. Data were analyzed using LongDistances v932 and the model-free fitting mode. The shaded error bands represent variation due to background noise, start time, dimensionality, and regularization. Histograms are predicted distances generated from the *in surfo* crystal structures 1NQG (blue) and 1NQH (pink) using the software package MMM (*Jeschke, 2018*).

*Figure 4 continued on next page*

*Figure 4 continued*

The online version of this article includes the following figure supplement(s) for figure 4:

**Figure supplement 1.** Raw double electron-electron resonance (DEER) data for the V90R1-T188R1 and S93R1-T188R1 spin pairs purified and reconstituted into 1-palmitoyl-2-oleoyl-glycero-3-phosphocholine (POPC).

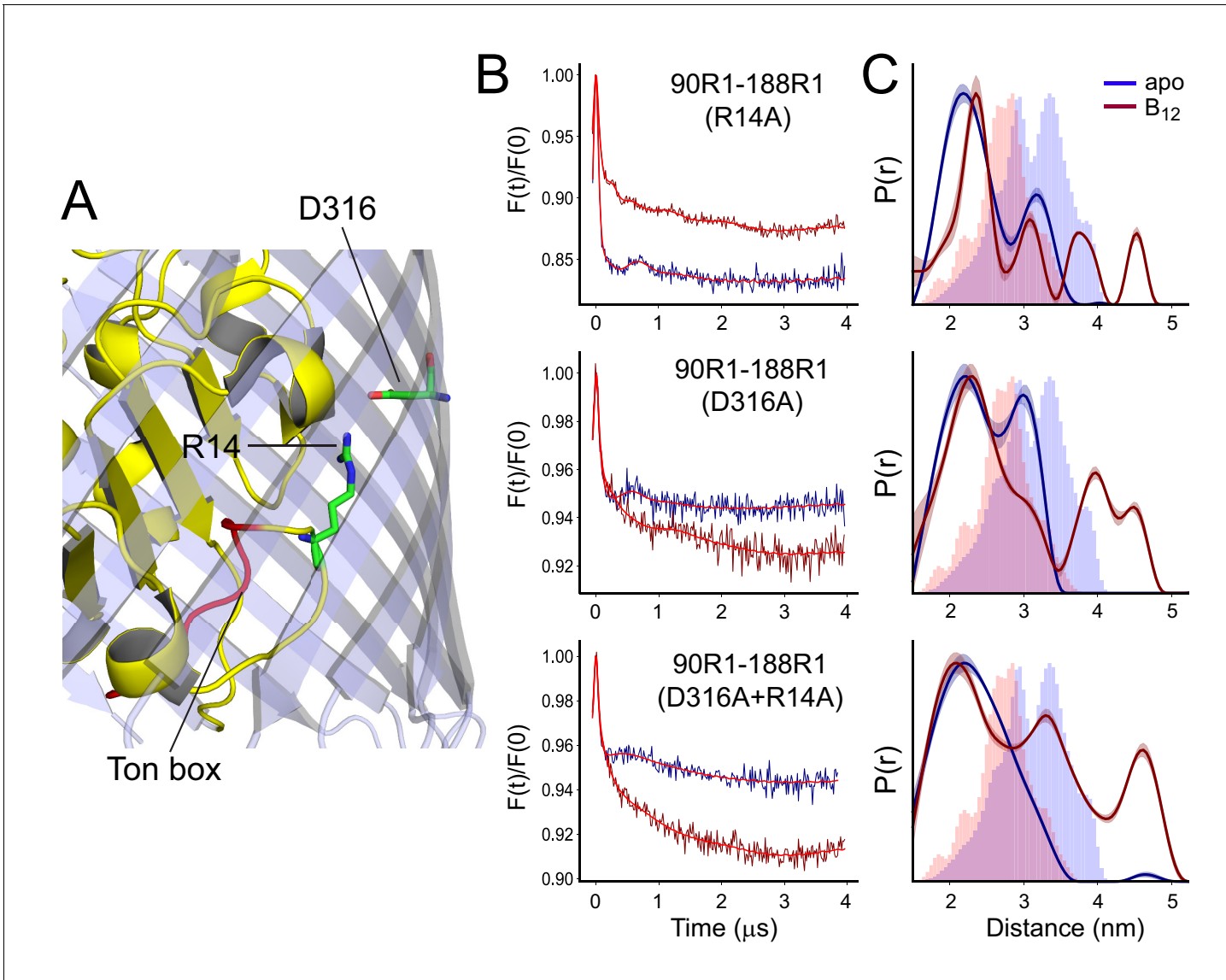

**Figure 5.** R14A, D316A, or D316A-R14A have similar effects on the conformation of SB3. In (A) is shown the structure of BtuB highlighting the positions of the R14 and D316 side chains and the location of the Ton box (from PDB ID: 1NQG). Background corrected double electron-electron resonance (DEER) data are shown in (B) and distance distributions in (C) for the V90R1-T188R1 spin pair in whole cells in the presence of the R14A, D316A, or the combined R14A-D316A mutants. Data are shown for both apo (blue) and vitamin B$_{12}$ bound (red) states. Lines through the DEER data represent the fits for the distributions shown on the right. These data were analyzed using LongDistances v932 and the model-free fitting mode. The shaded error bands in (C) represent variation due to background noise, start time, dimensionality, and regularization. Histograms represent predicted distances generated from the *in surfo* crystal structures for PDB ID: 1NQG (light blue) and PDB ID: 1NQH (pink) using the software package MMM (*Jeschke, 2018*).

The online version of this article includes the following figure supplement(s) for figure 5:

**Figure supplement 1.** Raw double electron-electron resonance (DEER) data for the V90R1-T188R1 spin pair in the presence of the R14A, D316A, and R14A/D316A mutations.

## Discussion

The pulse EPR measurements made here in intact *E. coli* indicate that SB3, which includes residues 82–96 in the core of BtuB, undergoes a substrate-induced structural change. When the ionic interaction between R14 in the core and D316 in the barrel is broken, an alternate and more dramatic structural change in SB3 occurs upon substrate binding, where SB3 is displaced approximately 2 nm toward the periplasmic surface of BtuB. Remarkably, these structural changes do not occur when the protein is removed from the native cell environment and reconstituted into a phospholipid bilayer, indicating that features in the intact cell environment modulate the energetics of the conformational states in BtuB.

Earlier experimental work provides evidence for an allosteric coupling between the substrate binding site, the R14-D316 ion pair, and the Ton box in BtuB. Measurements made by EPR in isolated OM or reconstituted phospholipid membranes demonstrated that substrate binding partially unfolded the Ton box (*Xu et al., 2006*; *Merianos et al., 2000*), and shifted the energy of the folded and unfolded Ton box states by about 2 kcal/mol (*Freed et al., 2010*). When the ionic interaction between R14 in the core and D316 in the barrel was broken, the Ton box was also observed to unfold and the coupling between substrate binding and the Ton box was broken (*Lukasik et al., 2007*). In addition, a connection between the Ton box and SB3 was seen by scintillation proximity assays where both the Ton box and SB3 were found to be necessary for a TonB-dependent retention of vitamin $B_{12}$ (*Mills et al., 2016*).

The connection between these sites in BtuB is also suggested by computational studies. When LPS is included in MD simulations, the interaction between R14 and D316 is weakened and the energy to unfold the Ton box reduced (*Balusek and Gumbart, 2016*). The inclusion of LPS also alters the state of SB3. In a symmetric phospholipid bilayer, SB3 assumes the more helical form, whereas in an asymmetric membrane containing LPS, SB3 assumes an extended form. These simulations are consistent with the results presented here, except that fully populating the extended form of SB3 in our whole cell measurement (state 2 in *Figure 2d*) requires substrate binding. Although the role of periplasmic components such as the peptidoglycan cannot be ruled out, the computational results indicate that interactions made by LPS with the extracellular loops of BtuB may alter conformational equilibria in the protein and provide an explanation for the differences in the behavior of BtuB when EPR measurements are made in cells versus reconstituted phospholipid bilayers. It should be noted that the interconversion between helical and extended forms for SB3 was also absent or diminished when the V90R1-T188R1 spin pair was examined by EPR in an isolated OM preparation (*Nilaweera et al., 2019*); as a result, the procedure to produce this OM preparation, which includes the use of sarkosyl, is apparently sufficient to modify the behavior of the protein.

An unexpected observation made here is that mutation of the R14-D316 ion pair alters the structure of the SB3 loop (*Figure 3D* and *Figure 5*). In the apo state of BtuB, this mutation enhances the extended form of SB3 (state 2 in *Figure 2B*), and upon substrate binding an alternate conformational state is generated where sites 90 and 93 are extended as much as 4.5 nm from site 188 on the 3/4 extracellular loop. Among the core sites examined, this more dramatic structural change involves only SB3 as labels in the first and second substrate binding loops (SB1 and SB2) at sites 63, 65, and 72 do not exhibit any significant structural changes. Such a large structural change involving SB3 has not been previously observed, and it appears to be localized to the C-terminal side of the core.

A high-resolution model obtained by crystallography for a fragment of TonB in complex with BtuB (*Shultis et al., 2006*) shows TonB interacting with the Ton box in an edge-to-edge manner (*Figure 6A*). Since the core is largely unaltered by TonB binding, models for transport have focused on the idea that TonB alters the core structure by exerting a mechanical force on the Ton box. In particular, TonB has been proposed to function by pulling on the Ton box, which then results in an unfolding of the N-terminal region of the core (*Hickman et al., 2017*). Single-molecule pulling experiments (*Hickman et al., 2017*) as well as steered MD simulations (*Gumbart et al., 2007*) suggest that pulling the Ton box will extract the N-terminal side of the core and eventually open a pore sufficient to allow substrate to pass. One difficulty with this model is that both experimental and computational approaches indicate that the extraction of an extended polypeptide chain longer than the width of the periplasm is required to open this pore.

We do not observe any significant structural changes in the N-terminal side of the core in cells (sites 63, 65, and 72), suggesting that the N-terminus may not move during transport. Rather, a

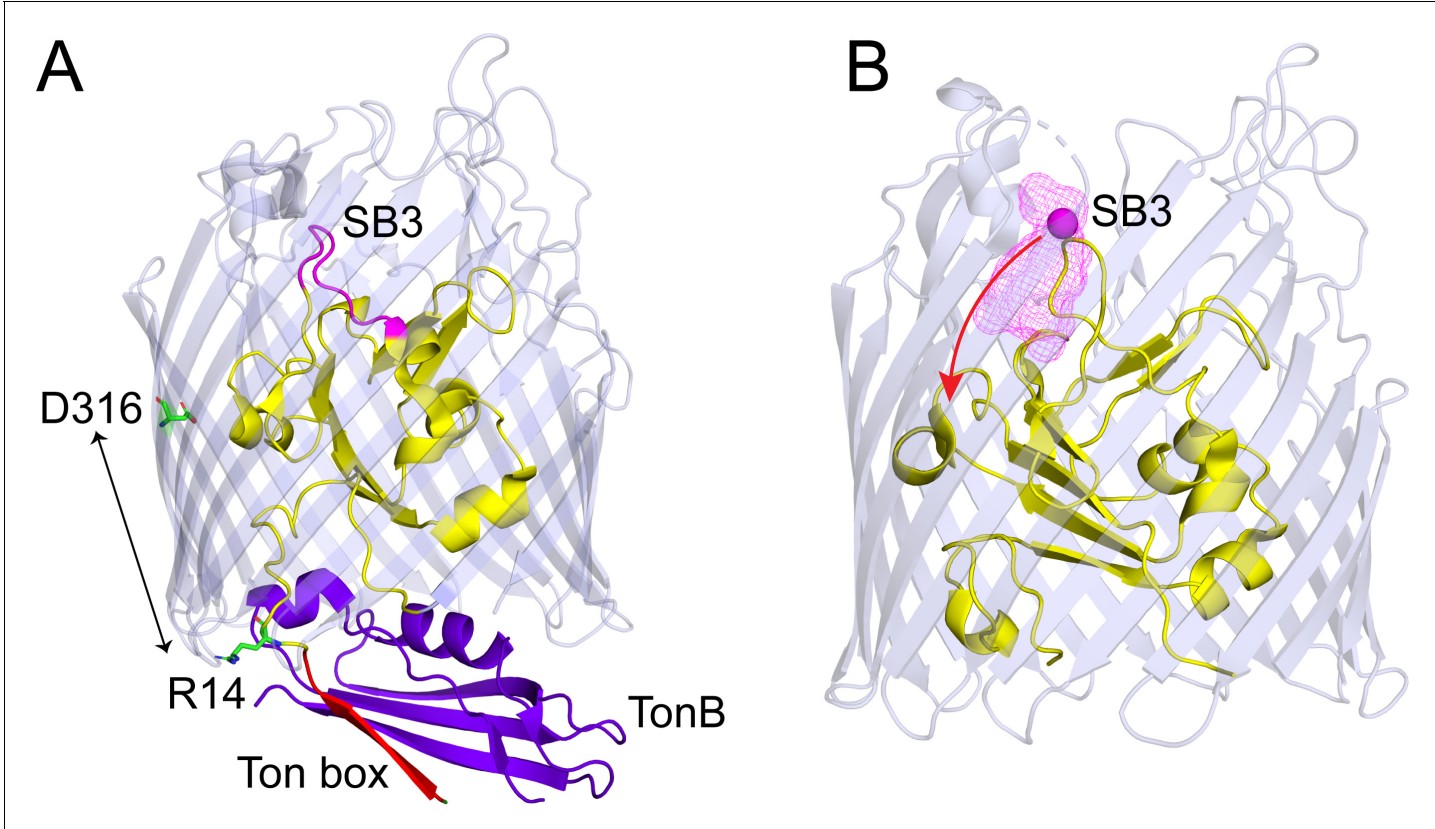

**Figure 6.** Conformational shifts in substrate binding loop 3 (SB3) with release of R14-D316 ionic lock. (**A**) View of BtuB (PDB ID: 2GSK) in complex with the C-terminal domain of TonB (purple) showing the core (yellow) and barrel (light blue) with the substrate binding loop SB3 (magenta), and the Ton box (red), where the R14-D316 ion pair has been broken. (**B**) An Xplor-NIH simulation showing positions of the Cβ carbon (magenta mesh) on site 90 in SB3 for 300 structures that are consistent with the distributions obtained by double electron-electron resonance (DEER) for V90R1-T188R1-R14A and V90R1-S237R1-R14A in the presence of vitamin $B_{12}$ (see Materials and methods). TonB binding extracts the Ton box from the core to create an edge-to-edge interaction with TonB, thereby breaking the R14-D316 ionic interaction. Breaking this interaction promotes the movement of SB3 toward the periplasmic interface of the transporter (red arrow) and may facilitate passage of vitamin $B_{12}$.

significant substrate-induced structural change is found to take place in SB3 on the C-terminal side of the core when the D316-R14 ionic lock is broken. At present, we have limited restraints to generate a model and we do not know the positions of many segments in the core, but a movement in SB3 that satisfies the EPR restraints is shown in *Figure 6B*. In this model, substrate binding moves SB3 into the barrel and toward the periplasm. We do not presently know whether the movement of SB3 is accompanied by the movement of substrate; however, it is interesting to note that the C-terminal side of the core has a lower side chain density than the N-terminal side, suggesting that there is more space for structural rearrangements within this region.

Presently, the precise sequence of steps that take place during transport are not known; however, both substrate and TonB are expected to be bound to BtuB at some point during transport, which should break the ionic lock between D316 and R14 (*Figure 6A*). This suggests that the large substrate-induced structural change observed here on the C-terminal side of the core (*Figure 6B*) will occur during transport. The energy to promote this structural change and disrupt core-barrel electrostatic interactions could be provided by the free energy of binding of TonB to BtuB, which is significant and characterized by a Kd in the nM range (*Freed et al., 2013*). To complete the transport cycle, TonB must be disengaged from BtuB and the inner membrane complex of ExbB and ExbD may perform this function and restore the apo state. Whether this involves a mechanical action of TonB, such as a pulling or rotational motion, or another process such as the exchange of the strand-to-strand TonB-Ton box interaction for a strand-to-strand interaction within a TonB dimer (*Freed et al., 2013*; *Gresock et al., 2015*) remains to be determined.

In summary, EPR spectroscopy in whole cells provides evidence for a substrate-induced structural transition in BtuB involving SB3 on extracellular apex of the core. The introduction of a mutation to break the R14-D316 ionic lock acting between the core and the barrel of BtuB produces an alternate structural state upon substrate addition so that SB3 is displaced as much as 2 nm into the barrel toward the periplasmic side of the protein. Under these same conditions, no movement of the N-terminal side of the core is detected. This ionic lock will be broken upon TonB binding and mutating this ionic lock may partially mimic the TonB bound state. As a result, this substrate-induced structural transition likely represents a structural state that occurs during the transport process. Remarkably, when BtuB is reconstituted into a phospholipid bilayer, these structural changes in SB3 are no longer observed, indicating that features in the native OM environment, such as the LPS, are required to populate conformational states that are important for BtuB function.

# Materials and methods

### Key resources table

| Reagent type (species) or resource | Designation | Source or reference | Identifiers | Additional information |
|---|---|---|---|---|
| Strain, strain background (*Escherichia coli*) | RK5016 (A derivative of MC4100 with the genotype *araD139 Δ(argF-lac)169 flbB5301 ptsF25 relA1 rpsL150 rbsR22 deoC1 gyrA219 non-9 metE70 argH1 btuB461 recA56*) | Robert Kadner (University of Virginia) | PMID:2982793 | *E. coli* strain lacking chromosomal gene for BtuB This strain was authenticated using phenotype assays |
| Strain, strain background (*Escherichia coli*) | RI90 (*araD139 Δ(araABC-leu)7679 galU galK Δ(lac)X74 rpsL thi phoR Δara714 leu+, dsbA:: Kanr*) | Coli Genetic Stock Center (Yale University, New Haven, CT) | PMID:8917542 | *E. coli* DsbA null strain This strain was authenticated using phenotype assays |
| Recombinant DNA reagent | pAG1 (plasmid) | Robert Kadner (University of Virginia) | pUC8 with btuB ORF (2.4 kb) and regulatory region | Plasmid containing WT BtuB gene |
| Recombinant DNA reagent | pAG1 with single point mutations in BtuB (L63C, S65C, N72C, S93C and S237C) | Applied Biological Materials (Richmond, BC, Canada) | | Plasmids used to construct and express BtuB with single mutations |
| Sequence-based reagent | BtuB D316A-FP | This paper | PCR primers | (5' – 3') GGTGCGGGTGTCGCCTG GCAGAAACAGACTAC |
| Sequence-based reagent | BtuB D316A-RP | This paper | PCR primers | (5' – 3') GTAGTCTGTTTCTGCC AGGCGACACCCGCACC |
| Sequence-based reagent | BtuB R14A-FP | This paper | PCR primers | (5' – 3') GTTACTGCTAACGCT TTTGAACAGCCGCGCA |
| Sequence-based reagent | BtuB R14A-RP | This paper | PCR primers | (5' – 3') TGCGCGGCTGTTCA AAAGCGTTAGCAGTAAC |
| Sequence-based reagent | BtuB V90C-FP | Nilaweera et al., Biophys. J. 117, 1476–1484. PMID:31582182 | PCR primers | (5' – 3') GAATCTGGCGGGGTG TAGTGGTTCTGCCG |
| Sequence-based reagent | BtuB V90C-RP | Nilaweera et al., Biophys. J. 117, 1476–1484. PMID:31582182 | PCR primers | (5' – 3') CGGCAGAACCACTA CACCCCGCCAGATTC |

*Continued on next page*

*Continued*

| Reagent type (species) or resource | Designation | Source or reference | Identifiers | Additional information |
|---|---|---|---|---|
| Sequence-based reagent | BtuB T188C-FP | Nilaweera et al., Biophys. J. 117, 1476–1484. PMID:31582182 | PCR primers | (5′ – 3′) ACCGGATGCCAAGC GCAGACAGATAACGATGG |
| Sequence-based reagent | BtuB T188C-RP | Nilaweera et al., Biophys. J. 117, 1476–1484. PMID:31582182 | PCR primers | (5′ – 3′) GCGCTTGGCATCCGG TATTACCATAGGCAACAAC |
| Chemical compound, drug | OG (octylglucoside or *n*-octyl-β-D-glucopyranoside) | Chem-Impex, international (Wood Dale, IL) | Cat# 00234 | Detergent for BtuB reconstitution |
| Chemical compound, drug | Vitamin B$_{12}$ (CN-Cbl, Cyanocobalamin) | Sigma Aldrich | Cat# V2876 | Substrate for BtuB |
| Chemical compound, drug | 1-Palmitoyl-2-oleoyl-glycero-3-phosphocholine | Avanti Polar Lipids, (Alabaster, AL) | POPC Cat#8 50457 | Lipid used for membrane reconstitution of BtuB |
| Chemical compound, drug | (1-Oxy-2,2,5,5-tetramethylpyrrolinyl-3-methyl)methanethiosulfonate | Cayman Chemical, Ann Arbor Michigan | MTSSL Cat# 16463 | Reagent for spin labeling protein cysteine residues |
| Software, algorithm | LongDistances (v. 932) | Christian Altenbach (UCLA) | LabVIEW software routine for the analysis of pulse EPR data | Used to examine DEER data |
| Software, algorithm | DeerAnalysis (v. 2019) | Gunnar Jeschke (ETH Zürich) | MATLAB routine for the analysis of pulse EPR data | Used to examine DEER data |
| Software, algorithm | MMM (v. 2018.2) | Gunnar Jeschke (ETH Zürich) | MATLAB routine for the determination of spin label rotamers and predicted label-label distances | Used in this study to predict distance distributions from crystal structures and in silico BtuB structures for simulated annealing |
| Software, algorithm | Xplor-NIH (v. 3.2) | Charles Schwieters, Marius Clore (NIH, NIDDK) | Used for simulated annealing to generate structures consistent with DEER data | |
| Software, algorithm | MATLAB (v. 2020a) | MathWorks, Inc (Natick, MA) | Program needed to execute DeerAnalysis and MMM | |
| Software, algorithm | Pymol (v. 2.4.0a0) | Schrödinger, LLC (New York, NY) | Program for molecular graphics | |

## Cell lines and mutants

The pAG1 plasmid with WT *btuB* gene and the RK5016 strain (*araD139 Δ(argF-lac)169 flbB5301 ptsF25 relA1 rpsL150 rbsR22 deoC1 gyrA219 non-9 metE70 argH1 btuB461 recA56*) used for growth assays were kindly provided by late professor R Kadner, University of Virginia. The *E. coli dsbA* null (*dsbA-*) mutant strain, RI90 (*araD139 Δ(araABC-leu)7679 galU galK Δ(lac)X74 rpsL thi phoR Δara714 leu+, dsbA:: Kanr*) were obtained from the Coli Genetic Stock Center (Yale University, New Haven, CT). L63C, S65C, N72C, S93C, and S237C *btuB* mutants were custom produced by Applied Biological Materials Inc (Richmond, BC, Canada). The *btuB* mutants (L63C-T188C, S65C-T188C, N72C-T188C, V90C-T188C, V90C-S237C, S93C-T188C) with and without the R14A mutation, and V90C, V90C-T188C-D316A, and V90C-T188C-R14A-D316A were engineered using polymerase chain reaction (PCR) mutagenesis. The plasmids were confirmed by sequencing and were transformed into

dsbA⁻ cells. Glycerol stocks were prepared and stored at −80°C. The RK5016 strain was authenticated using phenotype assays as described previously (*Lathrop et al., 1995*). This strain fails to grow in minimal media (MM) that is not supplemented with methionine and vitamin $B_{12}$. The RI90 strain carried kanamycin resistance and lacks DsbA function (*Rietsch et al., 1996*). This cell line was authenticated by testing for kanamycin resistance and determining that cells were not able to oxidize pairs of cysteine residues that were expressed on the cell surface (*Nilaweera et al., 2019*).

## Whole cell sample preparation

 dsbA⁻ cells expressing V90C-T188C, L63C, S65C, N72C, V90C, and S93C BtuB were grown in MM supplemented with 200 µg/ml ampicillin, 0.2 % w/v glucose, 150 µM thiamine, 3 mM $MgSO_4$, 300 µM $CaCl_2$, 0.01 % w/v methionine, and 0.01% w/v arginine (*Nilaweera et al., 2019*). Cells expressing BtuB with the V90C-T188C mutation was spin labeled as described (*Nilaweera et al., 2019*) and the aliquots of processed cell pellets were mixed with vitamin $B_{12}$ (0, 1, 5, 20, 30, 60, and 100 µM final concentrations). The cells expressing L63C, S65C, N72C, V90C, and S93C BtuB mutants were processed as described in *Nyenhuis et al., 2020b*.

Glycerol stocks of dsbA⁻ cells expressing L63C-T188C, S65C-T188C, N72C-T188C, V90C-T188C, S93C-T188C, and V90C-S237C BtuB with and without R14A, V90C-T188C-D316A, and V90C-T188C-R14A-D316A BtuB were used to directly inoculate the pre-precultures (Luria Bertani media with 200 µg/ml ampicillin), grown for 8 hr at 37°C and used to inoculate the MM precultures. The main MM cultures were inoculated with precultures, grown until $OD_{600}$ ~0.3, and spin labeled (*Nyenhuis et al., 2020b*). Briefly, the cells were spin labeled with methanethiosulfonate spin label (MTSSL) ((1-oxy-2,2,5,5-tetramethylpyrrolinyl-3-methyl)methanethiosulfonate) (Cayman Chemical, Ann Arbor, MI) in 100 mM HEPES buffer (pH 7.0) containing 2.5% (w/v) glucose with the final concentration of 7.5 nmol/ml of cell culture at $OD_{600}$ 0.3 for 30 min at room temperature (RT). Spin labeled cells were washed by resuspending in 2.5% (w/v) glucose supplemented 100 mM HEPES buffers, first, at pH 7.0 and then, at pD 7.0. During the washing steps, cysteine double mutants without R14A were incubated for 15 min, while 2–5 min for mutants with R14A and 10 min for D316A and for R14A-D316A mutants. For the data shown in *Figure 3—figure supplement 4*, aliquots of processed V90C-T188C-R14A samples were incubated with vitamin $B_{12}$ for 0, 30, and 60 min at RT. All other samples were incubated with vitamin $B_{12}$ for 20 min or less prior to freezing. It should be noted that previous work demonstrated that BtuB can be specifically labeled in these dsbA⁻ cells and that significant labeling of other OM proteins does not occur (*Nilaweera et al., 2019*).

## Reconstituted BtuB sample preparation

MM main cultures of V90C-T188C and S93C-T188C with and without R14A were grown for 8 hr at 37°C. The harvested cells were used to isolate intact OM (*Nyenhuis et al., 2020b*). After the second spin at 118,370 × g for 60 min at 4°C, the pellets were resuspended in 5 ml of HEPES buffer. The OMs were solubilized in 100 mM Tris pH 8.0 buffer with 10 mM EDTA and 0.5 g of octylglucoside (OG) (Chem-Impex, Wood Dale, IL). The OM suspension was incubated at 37°C for 10 min and 2 hr at RT and then spun at 64157 × g, 60 min at 4°C. The supernatants were used to spin label BtuB with 12 mM MTSSL at RT, overnight. BtuB was purified using six column volumes (CV) of wash buffer (17 mM OG, 25 mM Tris pH 8.0), 12 CV of 0–100% gradient of elution buffer (1 M NaCl, 17 mM OG, 25 mM Tris pH 8.0) and 6 CV of 100% elution buffer using a Q column and fractions containing BtuB were pooled. POPC (Avanti Polar Lipids, Alabaster, AL) (20 mg/ml) was sonicated in reconstitution buffer (150 mM NaCl, 100 nM EDTA, 10 mM HEPES pH 6.5) with OG (100 mg/ml) until clear, next, 1 ml from micelles mixture was added to each pooled BtuB mutant and incubated at RT, 40 min. BtuB was reconstituted into POPC by dialyzing OG over six buffer exchanges using reconstitution buffer and bio-beads (with minimum of 6 hr dialysis per exchange). Reconstituted BtuB was pelleted by centrifugation at 23425 × g for 40 min at 4°C, resuspended in 200 µl of reconstituted buffer and further concentrated to 50 µl by using Beckman airfuge. The samples were frozen and stored at −80°C.

## EPR spectroscopy

For CW EPR, 6 µl of cell pellet, or 6 µl of cell pellet with 100 µM vitamin $B_{12}$ were loaded into glass capillaries (0.84 OD, VitroCom, Mountain Lakes, NJ). Capillaries were loaded into a Bruker ER 4123D dielectric resonator (Bruker BioSpin, Billerica, MA) mounted to a Bruker EMX spectrometer.

Data were taken at X-band and at temperature with 100 G sweep width, 1 G modulation, and 2 mW of incident microwave power. EPR spectra on live *E. coli* that did not produce a signal-to-noise ratio of 7–10 or greater with a single 20 s field sweep failed to produce pulse echoes at Q-band pulse of adequate amplitude and were discarded. For pulse EPR, 16 µl of cell pellet, 20% glycerol, and 100 µM CNCbl (vitamin B$_{12}$) when applicable were combined and loaded into quartz capillaries (1.6 mm OD., VitroCom, Mountain Lakes, NJ). Samples were flash-frozen in liquid nitrogen and loaded into an EN5107D2 resonator (Bruker BioSpin, Billerica, MA). Data were collected on a Bruker E580 at Q-band and 50 K using a 300 W TWT Amplifier (Applied Systems Engineering, Benbrook, TX). The dead-time free 4-pulse DEER sequence was used for all experiments, with rectangular pulses of typical lengths π/2 = 10 ns and π = 20 ns, and a 75 MHz frequency separation. It should be noted that modulation depths obtained from whole cell samples were generally highly variable. We suspect that this is a result of the cells actively metabolizing during the labeling and washing steps in their preparation. Labeled whole cell samples that did not label well or produce sufficiently large modulation depths (approximately 4%) were re-grown and re-labeled. For the instrument settings and specific resonator used in this work, the modulation depth for a well-labeled protein is approximately 20%. Based upon this, labeling efficiencies for most samples were estimated to be approximately 40–60%.

## Data processing

CW EPR spectra were normalized by dividing by the spectral second integral using in-house python scripts. All pulse EPR data, except for data in *Figure 3—figure supplement 2*, were processed using LongDistances v 932 (Christian Altenbach, UCLA). Data were fit to a variable dimension background, after which the model-free mode was used for distance fitting. The value of the smoothing parameter was selected based on the L-curve, ensuring that the selected value passed through the major oscillations present in the data. Error analysis used 100 variations at the default values for background noise, start time, dimensionality, and regularization error. For data in *Figure 2d*, the data were instead fit using a model-based mode with two-Gaussian components with free position, width, and amplitude to investigate the dosage dependence of the substrate-dependent shift toward a shorter distance component for the 90–188 pair. *Supplementary file 1* contains the error analysis for these data. Data in *Figure 3—figure supplement 2* were processed using the DeerNet routine (*Worswick et al., 2018*) in DeerAnalysis (*Jeschke et al., 2006*). A folder of all Source Data (raw, unprocessed DEER data) has been provided. All EPR figures were generated using python scripts and the matplotlib plotting library. Protein structure images were generated using Pymol (*DeLano, 2002*). Simulated distance distributions were generated using the software package MMM v. 2018.2 and the default rotamer library (*Jeschke, 2018*; *Polyhach et al., 2011*).

## Modeling

The 90–188 and 90–237 distributions with the 14A mutation and in the presence of vitamin B$_{12}$ were used in the generation of a model (*Figure 6b*) for the motion of the apical hatch loop using the software package Xplor-NIH (v. 3.2). The starting structure for modeling was the *in surfo* structure crystal structure in the presence of cobalamin (PDB ID: 1NQH), which we previously determined to be closest to the native state of the extracellular loops in the native environment (*Nyenhuis et al., 2020a*). In that work, we found minimal evidence for motion of the extracellular loops, and we assumed that motion was localized to the SB3 loop. The template structure was labeled in silico with the R1 side chain at the 90, 188, and 237 positions using the software package MMM (v. 2017.2) and the default rotamer library. The top three rotamers were selected from the in silico labeling and used to generate three input structures for ensemble calculations, with the relative weights of the three rotamers conserved from the in silico labeling calculation.

During the calculation, the reference R1 sites in the barrel (188 and 237) were held entirely fixed. The R1 side chain at site 90, the underlying SB3 loop element comprising residues 81–104, and the adjoining, unstructured hatch region comprising residues 112–124 were fully mobile during runs, while all remaining residues had fixed backbone atoms and mobile side chains. The standard Xplor potentials BOND, ANGL, and IMPR were used in conjunction with the torsionDB and repel potentials for all elements, and DEER restraints were encoded as square well potentials using the noePot potential term. All potential terms were ensemble averaged across the three input structures. The

peak positions used in the modeling were the peak centered at 4.5 nm for 90–188, and the peak at 2.8 nm for 90–237. Full peak widths were used, with the square well stopping at 5% of maximum intensity. Randomization was introduced to the calculations using the randomizeTorsions function on the starting side chain positions of the mobile hatch elements. Following this, 10 rounds of 400 step Powell minimization using all potentials and with the mobile hatch elements were used to satisfy the experimental restraints. Structure calculation was repeated until both restraints were within error.

## Acknowledgements

We would like to thank Viranga Wimalasiri (University of Virginia) and Dr Harris Bernstein (NIH, NIDDK) for technical support with the bacterial growth assays.

## Additional information

### Funding

| Funder | Grant reference number | Author |
|---|---|---|
| Office of Extramural Research, National Institutes of Health | NIGMS GM035215 | David S Cafiso |
| Office of Extramural Research, National Institutes of Health | NIGMS S10OD025149 | David S Cafiso |

The funders had no role in study design, data collection and interpretation, or the decision to submit the work for publication.

### Author contributions

Thushani D Nilaweera, Conceptualization, Resources, Formal analysis, Validation, Investigation, Methodology, Writing - review and editing; David A Nyenhuis, Conceptualization, Resources, Data curation, Software, Formal analysis, Validation, Investigation, Visualization, Methodology, Writing - original draft, Writing - review and editing; David S Cafiso, Conceptualization, Resources, Data curation, Supervision, Funding acquisition, Visualization, Methodology, Writing - original draft, Project administration, Writing - review and editing

### Author ORCIDs

David S Cafiso (iD) https://orcid.org/0000-0002-3813-8721

### Decision letter and Author response

Decision letter https://doi.org/10.7554/eLife.68548.sa1
Author response https://doi.org/10.7554/eLife.68548.sa2

## Additional files

### Supplementary files

• Source data 1. Unprocessed EPR data.

• Supplementary file 1. Table showing the results of fitting double electron-electron resonance (DEER) data for V90R1-T188R1 taken at increasing concentrations of substrate to a two-component model.

• Transparent reporting form

### Data availability

Raw unprocessed DEER data are available in a compressed folder called "SourceData". The Pymol session file used to produce Figure 6b is included as a supplementary file.

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
