## [Decision Letter]

**Acceptance summary:**

This is an interesting study that pushes forward the understanding of structural changes associated with the TonB-dependent BtuB transporter in native *E. coli* membranes. Once of the most intriguing aspects of this investigation, is that it highlights the importance, or necessity in this case, of studying membrane protein dynamics in native settings. The results further the investigation into the mechanism of this protein by linking the structural changes to a salt bridge that is linked to TonB interactions. In addition, the high quality of the EPR spectroscopy data presented, and further development of protein specific labeling in the cell, is expected to advance the ability to study other membrane proteins using similar approaches.

**Decision letter after peer review:**

Thank you for submitting your article "Structural intermediates observed only in intact *Escherichia coli* indicate a mechanism for TonB-dependent transport" for consideration by *eLife*. Your article has been reviewed by 3 peer reviewers, and the evaluation has been overseen by a Reviewing Editor and Olga Boudker as the Senior Editor. The following individuals involved in review of your submission have agreed to reveal their identity: Reza Dastvan (Reviewer #2); Phillip Klebba (Reviewer #3).

Essential revisions:

This is an interesting study that demonstrates the importance of investigating BtuB in the native *E. coli* outer membrane environment. Technologically, the DEER data presented were viewed as being of high quality, pushing forward the ability to report conformational changes of membrane proteins in cells. Therefore, this study was regarded as innovative and there was general enthusiasm about the potential to advance the membrane protein field. However, there were several concerns about the interpretation of these conformational changes as directly informing on the transporter mechanism, particularly due to the lack of labeling and functional controls. The complete recommendations by the reviewers have been included below and should be addressed. In particular, the following list describes the essential revisions needed.

1) Please provide quantification of the labeling yields for the various cysteine mutants in this study. In addition, address the question of reproducibility of spectra obtained from different cellular preparations.

2) In order to provide a better framework for interpreting the conformational changes observed, please include a functional assessment of the key mutations examined in this study either by uptake assays or other approach.

3) The proposed mechanism of the breakage of the ionic lock as a transport intermediate is speculative at this point. Please revise the paper to address other possible interpretations that have been raised by the reviewers as detailed below.

*Reviewer #1 (Recommendations for the authors):*

• The conformational change shown in Figure 6b is not detailed enough. How would such a conformational change affect the other parts of the core domain. Could a more detailed figure be provided?

• The LPS is proposed to be the main factor affecting the differences detected in vivo versus in vitro. Periplasmic components could also be involved, especially the presence of a functional Ton complex. It would be interesting to assess the influence of Ton inhibitors during these experiments. With uptake experiments, the authors could also probe the importance of LPS directly.

• Line 87: there is a third structure of a TBDT bound to TonB, the FoxA-TonB complex (pdb 6I97, PMID 31385808)

• Line 111: define/introduce SB3

• Legend of figure 1: there is a typo in the pdb id, should be 1NQH, not 1NHQ

• Figure 3d, 4th panel from top (90R1-188R1 R14A), what are the 2 marginal peaks at 4 and 5nm for the apo state (blue curve)? And why are these not present in figure 5c 1st panel?

• In the Materials and methods section, lines 563-564, only for the V90C-T188C-R14 samples is indicated a time frame of the experiments. What is the time frame for the other samples?

*Reviewer #2 (Recommendations for the authors):*

A number of points require clarification.

Details:

1. The introduction would profit from citing Figure 1b when different domains of the transporter are discussed.

2. Concentration of the utilized vitamin B12 is not mentioned in the paper.

3. p. 6, l. 111: Abbreviation SB3 is not introduced.

4. P. 7 (caption of Figure 1): 2Azz is not described in the paper.

5. P. 8 (Figure 2d and corresponding Figure S1): Error analysis or confidence bands are not shown for Figure 2d. In Figure S1-c separating the data and introducing offsets in addition to showing the fits to the data are really helpful.

6. P. 8, l. 173 (caption of Figure 2): "hatch domain" is used instead of "core domain".

7. P. 10 (Figure 3 and corresponding Figure S2): Unless the reviewer is missing a point, the provided distance distributions in Figure 3 for pairs 63-188, 63-188-14A, 65-188, 65-188-14A, 72-188, 72-188-14A, 93-188 does not correspond to the background-corrected DEER data in Figure S2. Even if the dimensionality of the background is presumed to be around 2 because of the two-dimensional distribution of the labeled proteins in the outer membrane or excluded volume is considered, these background corrected data correspond to distance distributions that include broad/long component(s) between 4-8 nm. For instance, in Figure 2b and 2c, these long-distance components are shown. The contribution of such components is higher in Figure 3 and even if the x-axes are between 1.5 to 5 nm, we should see some of those components. A clarification for this review is greatly appreciated.

8. P. 12 (Figure 4): For analyses of pairs 90-188(+B12), 93-188(apo), 93-188-R14A(+B12), please see point 7.

9. P. 12, l. 338: "But significantly, for neither spin pair are the longer substrate-induced shifts that 339 were seen in Figure 3d observed in the reconstituted system." The reviewer agrees with the statement if the distance distributions are correctly presented in Figure 4. Please see points 7 and 8.

10. P. 13. (Figure 5): In the case of DEER distance distribution for the pair 90-188-D316A-R14A(+B12) the error analysis seems a bit underestimated.

11. P. 14, l. 387: "… single broad peak centered around 2.4 nm in the apo state" The corresponding background-corrected DEER spectrum in Figure 5 consists of two components. Please see point 7.

12. P. 16, l. 459: "We do not observe any significant structural changes in the N-terminal side of the core in cells, suggesting that the N-terminus may not move during transport." Please be more specific on the region that is considered the N-terminal side of the core and the supporting experimental data.

*Reviewer #3 (Recommendations for the authors):*

Biophysical experiments on living cells face more challenges than in vitro studies, like the interactions of probes with other molecules than the target molecule, lower intensity probe responses that may undermine accurate data collection, and interference from cellular background signals. Besides those obstacles, the manuscript by Nilaweera et al., "Structural intermediates observed only in intact *Escherichia coli*…," attempts to simultaneously label pairs of Cys side chains, in relatively close physical proximity, with extrinsic nitroxide reagents. This framework requires, but the manuscript lacks, controls that confirm the specificity and extent of covalent modification. This deficiency leads to questions about the validity and interpretations of ensuing EPR observations. Secondly, the paper describes structural changes in BtuB when it binds vitamin B12, but whether the conformational changes are pertinent to its transport mechanism is clouded by the impact of the mutations on BtuB structure and physiology. Potential over-interpretation creeps into the title of the paper, "Structural intermediates observed only in intact *Escherichia coli*…": whether the reorientation of SB3 that occurs in the mutant proteins during B12 binding constitutes a structural intermediate is debatable. Although the authors interpret their data as if they are structural coordinates, the resolution of the ESR observations does not reveal the exact magnitude, overall nature nor direction of probe motion. Protein dynamics are expected during ligand binding, as seen in many proteins, like the closely related TBDT FepA during its binding of ferric enterobactin. Loop motion probably also occurs in the BtuB-B12 binding reaction, but I'm not sure that the motion the authors describe has biological relevance.

1. In both this manuscript, and its predecessor (Nilaweera et al., 2018) the extent and stoichiometry of the MTSL labeling reactions with Cys mutant BtuB proteins are unknown. Because the authors do not provide these data it's difficult to compare one set of Cys pairs to another. To study BtuB in living bacteria the observed EPR spectra must originate from the target protein, in this case BtuB-MTSL. Confirmation requires evaluation of specific (BtuB) and non-specific (other proteins or cell components) labeling, as by western blots with anti-MTSL sera. The manuscript lacks these data, and refers to Nilaweera et al., 2018, that also lacks them. The referenced paper provides CW X-band spectra of cells expressing wild type BtuB or Cys-pair mutant derivatives, but it's not the same parameter. The problem is that the Cys mutant pairs spontaneously form disulfide bonds in vivo, unless the authors inactivate the DsbAB system (that normally facilitates disulfide formation). In this genetic background other cell envelope proteins become susceptible to modification by MTSL. It's relevant that other TBDT contain disulfide bonds in their surface loops, including Fiu (1), FepA (1), FhuA (2) and Cir (1), that are all expressed at high levels when *E. coli* is grown in minimal media. Besides this uncertainty about the extent of background labeling, the authors tacitly assume that each Cys side chain in each mutant pair is quantitatively labeled, but this may not be the case. One site in a pair may be 90% labeled, and the other 10% labeled. Without knowledge of the labeling distribution, it's hard to interpret the in vivo DEER data. It may be that all of the Cys sulfhydryls in all of the Cys pairs are equivalently quantitatively labeled, but my experience with Cys mutants suggest that this is not the case, and should be evaluated by experiments. I suggest that the authors quantify the MTSL attached to each Cys sulfhydryl of the 5 BtuB Cys pair mutants (e.g., T188C-V90C), and show the background labeling of other cell envelope proteins in each case. This is the only way to impart confidence about the origin of the EPR spectra. These additional data may resolve an inconsistency in the manuscript: CW data on V90C indicate its environment does not change when B12 binds (Figure 1), whereas DEER data on S188C – V90C pair suggest that MTSL attached to V90C relocates 25 Å when B12 binds (Figures 2 and 3). The authors do not address this discrepancy, but misunderstanding of the DEER data is a potential explanation. The authors also encountered a lot of labeling variability. About X-band spectra they stated: "…EPR spectra on live *E. coli* that did not produce a signal-to-noise ratio of 7 to 10 or greater with a single 20 second field sweep failed to produce pulse echoes at Q-band pulse of adequate amplitude and were discarded." For pulse EPR they encountered similar labeling problems: "… modulation depths obtained from whole cell samples were generally highly variable." Despite the rationalization "… this is a result of the cells actively metabolizing during the labeling and washing steps in their preparation," all the cell samples were presumably identically grown and processed, so what is the basis for the discrepancies? The problem likely originates from irreproducibility in the MTSL labeling reactions. Statistical analyses of the different trials are needed to resolve the variability. It's also necessary to know the exact number of cells that were loaded for CW, and frozen and analyzed by pulsed EPR. Were equivalent amounts of all spin labeled samples used for each CW or pulsed EPR experiments? Because the authors do not mention nor experimentally address these issues, conclusions about the mobility or proximity of the extrinsic probes may be unreliable.

2. The effects of the R14A mutation are interpreted to support the contention of mechanistic relevance, but they may be engendered by the mutation itself. The R14A substitution excludes ionic interactions with D316, but it also substitutes one of the smallest amino acid side chains in place of one of the largest. Besides that, X-ray data show that cobalamin directly contacts both V90 and S93 in SB3 when it binds to BtuB. The attachment of MTSL to these sites adds non-relevant bulk to the surface of the binding pocket. The combination of removing bulk from the BtuB interior, and adding bulk to the SB3 loop directly above it, may impact the tertiary structure of SB3 during the BtuB-B12 binding interaction. Just eliminating the proposed association between R14 and D316 (as in D316A) may similarly destabilize the protein, ultimately altering both affinity for B12 and the structural properties of the OM protein. It seems possible that R14A changes aspects of the binding complex at equilibrium, resulting in unnatural movement of V90 and S93 in unorthodox directions. The proposed movement of the loop toward the periplasm (the actual direction of the motion is not known from the reported data) is consistent with this alternative explanation. The authors previously showed that mutational alterations at R14 or D316 disrupt the structure of the BtuB core, especially with regard to the TonB box region, directly beneath SB3 (Lukasik et al., 2007). Those data originated from purified BtuB Cys mutants, labeled with MTSL. In the current manuscript they observed that the same mutations also affected the configuration of SB3 when B12 bound. In this case the conformational changes were only observed in native outer membranes, for unknown reasons that the authors attribute to the effects of LPS. As a result of these numerous possibilities, it's not certain that R14A or D316A create mechanistically relevant intermediates. It's also conceivable that the distortion of the core caused by the mutations has a domino effect on BtuB structure that ultimately results in aberrant, or novel localization of SB3 during B12 binding. So, the results do not necessarily define a transport reaction intermediate. If not, then neither do they provide mechanistic insight into TonB-dependent transport.

The lack of information about the phenotypes of the mutants themselves, whether it's R14A, D316A, or those that introduce modifiable Cys residues in 6 different positions, is a deficiency of this manuscript. I don't see how to make conclusions about the TBDT transport process without knowing more about the functionality of these constructs relative to native BtuB. What are the affinities of the mutant proteins for B12, and how does modification with MTSL of single and double Cys sulfhydryls affect binding and transport? The authors do not provide information about these key biochemical parameters; any or all of the mutants may be physiologically defective, and hence provide little or no insight into the transport reaction.

3. The authors sometimes misinterpret, misquote or neglect the literature. For example, they ignore the work that involved Jim Feix's lab, on SDSL of FepA 25 years ago, as well as other data on FepA, that demonstrated conformational motion akin to what they now describe for BtuB, both in vitro (Liu et al., 1995) and in vivo (Jiang et al., 1997). Their statement "…Computational work suggests.… that environment, specifically LPS, may be important in controlling the configuration of SB3" neglects other work on TBDT that demonstrated their completely different properties in vitro and in vivo. FepA has >100-fold lower affinity for ferric enterobactin when purified and reconstituted into detergent or liposomes (Payne et al., 1997), than when measured in bacterial cells (Newton et al., 1999). It's more difficult to rationalize an effect of LPS on SB3, that is ensconced with the 22-stranded β-barrel of BtuB, and thereby isolated from interactions with LPS.

---

## [Author Response]

Essential revisions:This is an interesting study that demonstrates the importance of investigating BtuB in the native *E. coli* outer membrane environment. Technologically, the DEER data presented were viewed as being of high quality, pushing forward the ability to report conformational changes of membrane proteins in cells. Therefore, this study was regarded as innovative and there was general enthusiasm about the potential to advance the membrane protein field. However, there were several concerns about the interpretation of these conformational changes as directly informing on the transporter mechanism, particularly due to the lack of labeling and functional controls. The complete recommendations by the reviewers have been included below and should be addressed. In particular, the following list describes the essential revisions needed.1) Please provide quantification of the labeling yields for the various cysteine mutants in this study. In addition, address the question of reproducibility of spectra obtained from different cellular preparations.

Quantitation of labeling yields. From the modulation depths in the DEER data, we provide a rough estimate of the efficiency of double labeling relative to the total spin count. Unlike a continuous wave EPR spectrum where a poorly labeled sample can yield usable data, the modulation depth in the DEER signal is dependent upon the fraction of spin-pairs excited. We now include an estimate of labeling efficiency, which typically ranges from 40 to 60% in our samples. Making a more precise measurement of double labeling efficiency in cells relative to the total BtuB count may not be possible; moreover, it is not clear why an exact labeling efficiency matters for the interpretation of these data? Lower or higher labeling efficiencies will alter the quality (signal-to-noise) of the data and affect the error in the distance distributions, but they will not change the result.

Specificity in labeling. The method used here produces labeling that is specific. Previous work using this method on DsbA- cells demonstrated that there was little labeling of the cells when expressing wild-type BtuB (which lacks cysteine) and that no DEER signal can be detected (see: Nilaweera et al. (2019) Biophysical journal 117, 1476-1484). Two previous publications also show DEER signals that arise from specifically labeled BtuB in DsbA- cells (Nyenhuis et al. (2020) JACS 142, 10715-10722; Nyenhuis et al. (2020) Biophys. J. 119, 1550-1557). And in the latter Biophys. J. publication, a wide range of distances is obtained that match well with the predictions from crystal structures.

Reproducibility of spectra. The modulation depths in the DEER data and signal levels from our preparations do vary, despite our best efforts to control for it. However, the EPR spectra are consistent, and DEER data are highly reproducible even when the labeling efficiencies vary.

2) In order to provide a better framework for interpreting the conformational changes observed, please include a functional assessment of the key mutations examined in this study either by uptake assays or other approach.

We performed a growth assay on the ionic mutant used it this study, and it shows that the protein is functional and not damaged in a significant way.

3) The proposed mechanism of the breakage of the ionic lock as a transport intermediate is speculative at this point. Please revise the paper to address other possible interpretations that have been raised by the reviewers as detailed below.

We revised the paper to focus just on what we believe the data tell us, and these changes are detailed below. The data suggest that substrate binding loop 3 undergoes a motion towards the periplasm during the transport cycle. We agree with the reviewers that the exact role of this structural transition, or whether it is necessarily directly involved in transport, is not known at this time.

Reviewer #1 (Recommendations for the authors):• The conformational change shown in Figure 6b is not detailed enough. How would such a conformational change affect the other parts of the core domain. Could a more detailed figure be provided?

We wanted to see what the DEER data implied regarding the movement of SB3, and Figure 6b is based upon Xplor-NIH modeling using the restraints obtained for the third substrate binding loop SB3. We found that this loop could move and satisfy the EPR restraints without significant rearrangements of the remainder of the core. This was not an MD simulation and we do not have enough data on other regions of the core to know how it might rearrange.

We have provided a Pymol session file showing 300 structures in SB3 that are consistent with the distance data. This should provide the reader with as much detail of this figure as is needed.

• The LPS is proposed to be the main factor affecting the differences detected in vivo versus in vitro. Periplasmic components could also be involved, especially the presence of a functional Ton complex. It would be interesting to assess the influence of Ton inhibitors during these experiments. With uptake experiments, the authors could also probe the importance of LPS directly.

The reviewer is correct, periplasmic components might also be important. However, there is not enough TonB present in these experiments relative to BtuB to account for the altered distributions seen in cell system. Nonetheless looking at the role of TonB directly is something we are attempting to carry out. Because other components such as peptidoglycan might be playing a role, we have amended our comments on the third paragraph in the Discussion to reflect this fact. As mentioned in the Discussion, we focused on LPS because there is already support from computational work for the role of LPS on the energetics of the BtuB core region (see: Balusek and Gumbart, Biophys. J. (2016) 111, 1409-1417).

• Line 87: there is a third structure of a TBDT bound to TonB, the FoxA-TonB complex (pdb 6I97, PMID 31385808)

We added a reference to this structure in the introduction.

• Line 111: define/introduce SB3.

We have now defined “SB3” in the introduction.

• Legend of figure 1: there is a typo in the pdb id, should be 1NQH, not 1NHQ.

The typo has been corrected.

• Figure 3d, 4th panel from top (90R1-188R1 R14A), what are the 2 marginal peaks at 4 and 5nm for the apo state (blue curve)? And why are these not present in figure 5c 1st panel?

These small peaks in the apo state for 90R1-188R1 R14A may indicate a very small incidence of the shifted state of SB3 even in the absence of cobalamin, which is not seen for reconstituted protein (as in Figure 5c). However, these peaks are small enough to preclude reliable interpretation, and so we were not comfortable drawing conclusions from them in the manuscript. Getting true error estimates on distance distributions obtained from DEER is a difficult problem due to the mathematical deconvolution of the dipolar evolution into specific frequency and distance components. Developing methods for error analysis in the processing of DEER data is currently a topic of some interest. The ranges provided in most of the current software packages give a range of solutions (distance distributions) relative to the best fits.

• In the Materials and methods section, lines 563-564, only for the V90C-T188C-R14 samples is indicated a time frame of the experiments. What is the time frame for the other samples?

The timeframe used for the other samples (from label addition to freezing for the DEER experiment) was typically 20 minutes or less, and we have now included this information in the methods.

Reviewer #2 (Recommendations for the authors):A number of points require clarification.Details:1. The introduction would profit from citing Figure 1b when different domains of the transporter are discussed.

We added a reference to Figure 1b in the introduction.

2. Concentration of the utilized vitamin B12 is not mentioned in the paper.

Unless otherwise mentioned, the vitamin B_12_ concentration was 100 μM. This is mentioned in the Methods section.

. p. 6, l. 111: Abbreviation SB3 is not introduced.

We have now defined this abbreviation in the introduction where it is first used.

4. P. 7 (caption of Figure 1): 2Azz is not described in the paper.

We re-wrote this caption to better indicate the points being measured and to make the reference to 2Azz clearer.

5. P. 8 (Figure 2d and corresponding Figure S1): Error analysis or confidence bands are not shown for Figure 2d. In Figure S1-c separating the data and introducing offsets in addition to showing the fits to the data are really helpful.

The program LongDistances treats error differently for a model-based fit than for a model-free fit. It allows for negative probabilities in the distributions, which generates unrealistic errors for data that have smaller modulation depths. However, the model-based fit does generate an error for each component, and we added a table (Supplementary File 1) which lists the proportions of the two distance components as a function of vitamin B_12_ as well as the error in the fits.

We modified Figure 2—figure supplement 1, to separate the data with offsets and show fits to the data.

6. P. 8, l. 173 (caption of Figure 2): "hatch domain" is used instead of "core domain".

We made the correction.

7. P. 10 (Figure 3 and corresponding Figure S2): Unless the reviewer is missing a point, the provided distance distributions in Figure 3 for pairs 63-188, 63-188-14A, 65-188, 65-188-14A, 72-188, 72-188-14A, 93-188 does not correspond to the background-corrected DEER data in Figure S2. Even if the dimensionality of the background is presumed to be around 2 because of the two-dimensional distribution of the labeled proteins in the outer membrane or excluded volume is considered, these background corrected data correspond to distance distributions that include broad/long component(s) between 4-8 nm. For instance, in Figure 2b and 2c, these long-distance components are shown. The contribution of such components is higher in Figure 3 and even if the x-axes are between 1.5 to 5 nm, we should see some of those components. A clarification for this review is greatly appreciated.

As the reviewer correctly notes, the difference between the background corrected DEER data and the distance distribution is due to a long-distance contribution. This contribution is in the 6 nm range, and it results from intermolecular BtuB-BtuB interactions due to the formation of what are termed “OMP islands.” The size of this interaction is variable, and it “pollutes” many of our DEER traces in the native system. As a result, we have truncated the distributions at 5 nm. We published a paper last year describing this intermolecular signal and used it to define the interfaces of BtuB that interact in the outer membrane (Nyenhuis et al. (2020) JACS, 142, 10715-10722).

To avoid confusing the reader who carefully looks at the data, we added a sentence to the paragraph that introduces Figure 3 to make the reader aware of this feature.

8. P. 12 (Figure 4): For analyses of pairs 90-188(+B12), 93-188(apo), 93-188-R14A(+B12), please see point 7.

See response to point 7.

9. P. 12, l. 338: "But significantly, for neither spin pair are the longer substrate-induced shifts that 339 were seen in Figure 3d observed in the reconstituted system." The reviewer agrees with the statement if the distance distributions are correctly presented in Figure 4. Please see points 7 and 8.

See response to point 7. Intermolecular BtuB-BtuB interactions generally do not appear in the reconstituted system, and unlike the intermolecular measurements made to SB3, they are not modulated by substrate.

10. P. 13. (Figure 5): In the case of DEER distance distribution for the pair 90-188-D316A-R14A(+B12) the error analysis seems a bit underestimated.

Yes, we agree with the reviewer that the error analysis seems to be underestimated for Figure 5. Nonetheless, this was the output of Christian Altenbach’s program LongDistances. The raw data is available for readers who wish to re-analyze the data or process the data in a different manner. It should be noted that with an echo time of 4 μsec, distances and distributions below 5 nm are expected to be reliable.

11. P. 14, l. 387: "… single broad peak centered around 2.4 nm in the apo state" The corresponding background-corrected DEER spectrum in Figure 5 consists of two components. Please see point 7.

As described above (response to point 7), the data reflects the presence of BtuB-BtuB interactions due to the formation of OMP islands. The effect of intermolecular BtuB-BtuB interactions does not appear past 5 nm. By truncating the distribution at this point, the remaining distribution represents the 90R1-188R1 interaction.

12. P. 16, l. 459: "We do not observe any significant structural changes in the N-terminal side of the core in cells, suggesting that the N-terminus may not move during transport." Please be more specific on the region that is considered the N-terminal side of the core and the supporting experimental data.

We are referring to measurements made to sites 63, 65 and 72 in SB1 and SB2. We modified the discussion to refer to these specific sites so that this is clear to the reader.

Reviewer #3 (Recommendations for the authors):Biophysical experiments on living cells face more challenges than in vitro studies, like the interactions of probes with other molecules than the target molecule, lower intensity probe responses that may undermine accurate data collection, and interference from cellular background signals. Besides those obstacles, the manuscript by Nilaweera et al., "Structural intermediates observed only in intact *Escherichia coli*…," attempts to simultaneously label pairs of Cys side chains, in relatively close physical proximity, with extrinsic nitroxide reagents. This framework requires, but the manuscript lacks, controls that confirm the specificity and extent of covalent modification. This deficiency leads to questions about the validity and interpretations of ensuing EPR observations. Secondly, the paper describes structural changes in BtuB when it binds vitamin B12, but whether the conformational changes are pertinent to its transport mechanism is clouded by the impact of the mutations on BtuB structure and physiology. Potential over-interpretation creeps into the title of the paper, "Structural intermediates observed only in intact *Escherichia coli*…": whether the reorientation of SB3 that occurs in the mutant proteins during B12 binding constitutes a structural intermediate is debatable. Although the authors interpret their data as if they are structural coordinates, the resolution of the ESR observations does not reveal the exact magnitude, overall nature nor direction of probe motion. Protein dynamics are expected during ligand binding, as seen in many proteins, like the closely related TBDT FepA during its binding of ferric enterobactin. Loop motion probably also occurs in the BtuB-B12 binding reaction, but I'm not sure that the motion the authors describe has biological relevance.

There are three points we would like to make regarding these comments. First, as detailed below, previously published work has demonstrated specificity in our labeling – it is the first control experiment we perform. We have also now provided relative estimates of labeling efficiency from the modulation depths in the DEER signal, which is now mentioned in the Methods section where the DEER experiment is described.

Second, we agree with the reviewer that we do not know that the structural transition that we observe in substrate binding loop 3 is related to transport. We have not fully characterized the BtuB core, and other motions may occur that we have not yet observed. All we can say is that the transition we see may occur during the transport process. To address this concern (as well similar concerns raised by the other reviewers), we have modified portions of the text and two paragraphs in the Discussion. Other than the partial unfolding of the very N-terminal segment of BtuB (the Ton box), this movement of SB3 is the only site in the core of BtuB that has been observed to undergo an extensive substrate-induced structural change.

Finally, we do not understand the comment that these EPR measurements do not reveal the exact nature and overall motion of the probe. This is precisely what these measurements do. They provide a measure of the interspin distances in the labeled protein. We demonstrated in an earlier work that the loops in BtuB are relatively static in the intact cell, as a result the movements seen for substrate binding loop 3 (SB3) can be entirely attributed the movement of this loop.

The reviewer may be interested to know that the loops in BtuB do not gate in the native cell environment as they do in a purified phospholipid environment. This may not be true for iron transporters like FepA, but at least for BtuB the loops are relatively static (Nyenhuis et al., Biophys. J. (2020) 119, 1550-1557).

1. In both this manuscript, and its predecessor (Nilaweera et al., 2018) the extent and stoichiometry of the MTSL labeling reactions with Cys mutant BtuB proteins are unknown. Because the authors do not provide these data it's difficult to compare one set of Cys pairs to another. To study BtuB in living bacteria the observed EPR spectra must originate from the target protein, in this case BtuB-MTSL. Confirmation requires evaluation of specific (BtuB) and non-specific (other proteins or cell components) labeling, as by western blots with anti-MTSL sera. The manuscript lacks these data, and refers to Nilaweera et al., 2018, that also lacks them. The referenced paper provides CW X-band spectra of cells expressing wild type BtuB or Cys-pair mutant derivatives, but it's not the same parameter. The problem is that the Cys mutant pairs spontaneously form disulfide bonds in vivo, unless the authors inactivate the DsbAB system (that normally facilitates disulfide formation). In this genetic background other cell envelope proteins become susceptible to modification by MTSL. It's relevant that other TBDT contain disulfide bonds in their surface loops, including Fiu (1), FepA (1), FhuA (2) and Cir (1), that are all expressed at high levels when *E. coli* is grown in minimal media.

We understand the reviewer’s concern, but this was one of our first control experiments and several published observations indicate that we are not getting significant labeling of proteins other than BtuB in the DsbA- strain. The reviewer mentions the Nilaweera et al. 2018 paper – we presume he is referring to our 2019 paper (Nilaweera et al. (2019) Biophysical J. 117, 1476-1484). First, the DsbA- strain without BtuB expression does not label to any significant extent (see Figure S4 – Nilaweera et al., 2019), and we cannot obtain a DEER signal because there is no echo upon which to set up the pulse experiment. The same is true with WT BtuB is expressed in the DsbA- or DsbB- strains (Figure 2, Nilaweera et al., 2019). Second, the CW EPR spectra we obtain from the DsbA- strain are consistent with our expectations based upon spectra measured from single labeled sites in the RK5016 strain or from purified reconstituted protein. Finally, the interspin distances, which we only observe when BtuB having double cysteines is overexpressed, matches that expected from the crystal structures (see for example Nyenhuis et al. (2020) Biophys. J. 119, 1550-1557).

To ensure that other readers to not miss this important point, we added a note addressing the specificity of labeling in the Methods section under the section “Whole cell sample preparation.”

Besides this uncertainty about the extent of background labeling, the authors tacitly assume that each Cys side chain in each mutant pair is quantitatively labeled, but this may not be the case. One site in a pair may be 90% labeled, and the other 10% labeled. Without knowledge of the labeling distribution, it's hard to interpret the in vivo DEER data. It may be that all of the Cys sulfhydryls in all of the Cys pairs are equivalently quantitatively labeled, but my experience with Cys mutants suggest that this is not the case, and should be evaluated by experiments. I suggest that the authors quantify the MTSL attached to each Cys sulfhydryl of the 5 BtuB Cys pair mutants (e.g., T188C-V90C), and show the background labeling of other cell envelope proteins in each case. This is the only way to impart confidence about the origin of the EPR spectra. These additional data may resolve an inconsistency in the manuscript: CW data on V90C indicate its environment does not change when B12 binds (Figure 1), whereas DEER data on S188C – V90C pair suggest that MTSL attached to V90C relocates 25 Å when B12 binds (Figures 2 and 3). The authors do not address this discrepancy, but misunderstanding of the DEER data is a potential explanation.

The reviewer is in error here. We do not assume or even need quantitative labeling of our cysteine sites. It is in fact not necessary to know the exact extent of labeling to correctly interpret the DEER data – much in the same way that it is not necessary to know labeling efficiency or even have matched donor/acceptor pairs on all proteins in a single molecule FRET measurement. All that is necessary is that sufficient sites be labeled so that an adequate number of spin pairs are excited in the 4-pulse DEER experiment, resulting in a measurable dipolar evolution with good signal-to-noise. Our maximum modulation depths (with our current probe and instrument configuration) are typically around 20% for well labeled protein (both sites near 100%). Taking this value, we can say that our double sites are typically labeled to an efficiency that range from about 40 to 60%. This is adequate but not as good as we would obtain for an in-vitro sample. Unlike a CW EPR spectrum where you might get good signals over quite a wide range of labeling efficiencies with sufficient signal averaging, the DEER signal is very sensitive to labeling efficiency.

We have included an estimate of labeling efficiency in the Methods. Information on the exact level of labeling efficiency in the intact cell is both unnecessary and very difficult to obtain.

Regarding the discrepancy where the label at site 90 shows no change in the EPR spectrum upon ligand binding, we do not agree that there is an inconsistency here. The environment around the label may change, but this does not necessarily mean that the EPR spectrum must change. In the present case, the label is at or near its rigid limit and only an increase in motion is likely to produce a change in EPR lineshape. A change in structure or environment that does not change the motion of the probe on the ns timescale is unlikely to significantly change the CW spectrum. Two labels buried in a protein interior will generally yield very similar if not identical EPR spectra. At the same time, the DEER data provide unequivocal evidence that the label position is changing.

The authors also encountered a lot of labeling variability. About X-band spectra they stated: '…EPR spectra on live *E. coli* that did not produce a signal-to-noise ratio of 7 to 10 or greater with a single 20 second field sweep failed to produce pulse echoes at Q-band pulse of adequate amplitude and were discarded." For pulse EPR they encountered similar labeling problems: "… modulation depths obtained from whole cell samples were generally highly variable." Despite the rationalization "… this is a result of the cells actively metabolizing during the labeling and washing steps in their preparation," all the cell samples were presumably identically grown and processed, so what is the basis for the discrepancies? The problem likely originates from irreproducibility in the MTSL labeling reactions. Statistical analyses of the different trials are needed to resolve the variability. It's also necessary to know the exact number of cells that were loaded for CW, and frozen and analyzed by pulsed EPR. Were equivalent amounts of all spin labeled samples used for each CW or pulsed EPR experiments? Because the authors do not mention nor experimentally address these issues, conclusions about the mobility or proximity of the extrinsic probes may be unreliable.

We were being quite transparent and honest in stating that not all our attempts at labeling yielded adequate signal-to-noise to produce reasonable DEER data. We were as reproducible as we could be, and we do not know all the reasons for the variability (perhaps variability in the riboswitch expression system intrinsic to BtuB). However, we fail to see why this is an issue. Unlike a CW spectrum, which might be quite easy to acquire over a range of concentrations given sufficient signal averaging, the pulse experiment is more sensitive to relative levels of labeled material and much more time consuming – particularly if labeled protein levels are not adequate. It is not clear what a “statistical analyses of different trials” would tell us or even how it would be carried out.

We do not understand the reviewer’s comment that our conclusions about probe mobility or proximity are unreliable. In every case where we have made a measurement under conditions where a known structure is expected, we get the expected result. We refer the reviewer to our most recent Biophysical Journal paper (Nyenhuis et al. (2020) Biophys. J. 119, 1550-1557) using the current approach where we make a wide range of inter-loop distance measurements. All these measurements yield distances that fall completely within the expectations of the in-surfo crystal structures. There is no evidence that distances we obtain are unreliable.

2. The effects of the R14A mutation are interpreted to support the contention of mechanistic relevance, but they may be engendered by the mutation itself. The R14A substitution excludes ionic interactions with D316, but it also substitutes one of the smallest amino acid side chains in place of one of the largest. Besides that, X-ray data show that cobalamin directly contacts both V90 and S93 in SB3 when it binds to BtuB. The attachment of MTSL to these sites adds non-relevant bulk to the surface of the binding pocket. The combination of removing bulk from the BtuB interior, and adding bulk to the SB3 loop directly above it, may impact the tertiary structure of SB3 during the BtuB-B12 binding interaction. Just eliminating the proposed association between R14 and D316 (as in D316A) may similarly destabilize the protein, ultimately altering both affinity for B12 and the structural properties of the OM protein. It seems possible that R14A changes aspects of the binding complex at equilibrium, resulting in unnatural movement of V90 and S93 in unorthodox directions. The proposed movement of the loop toward the periplasm (the actual direction of the motion is not known from the reported data) is consistent with this alternative explanation. The authors previously showed that mutational alterations at R14 or D316 disrupt the structure of the BtuB core, especially with regard to the TonB box region, directly beneath SB3 (Lukasik et al., 2007). Those data originated from purified BtuB Cys mutants, labeled with MTSL. In the current manuscript they observed that the same mutations also affected the configuration of SB3 when B12 bound. In this case the conformational changes were only observed in native outer membranes, for unknown reasons that the authors attribute to the effects of LPS. As a result of these numerous possibilities, it's not certain that R14A or D316A create mechanistically relevant intermediates. It's also conceivable that the distortion of the core caused by the mutations has a domino effect on BtuB structure that ultimately results in aberrant, or novel localization of SB3 during B12 binding. So, the results do not necessarily define a transport reaction intermediate. If not, then neither do they provide mechanistic insight into TonB-dependent transport.The lack of information about the phenotypes of the mutants themselves, whether it's R14A, D316A, or those that introduce modifiable Cys residues in 6 different positions, is a deficiency of this manuscript. I don't see how to make conclusions about the TBDT transport process without knowing more about the functionality of these constructs relative to native BtuB. What are the affinities of the mutant proteins for B12, and how does modification with MTSL of single and double Cys sulfhydryls affect binding and transport? The authors do not provide information about these key biochemical parameters; any or all of the mutants may be physiologically defective, and hence provide little or no insight into the transport reaction.

The reviewer raises important points regarding the consequences of the mutations and the incorporation of spin labels into BtuB. First, regarding the ionic interactions surrounding R14 and D316, the reviewer is concerned that the R14A mutation might destabilize the protein and induce unnatural structural rearrangements. An inspection of the TonB-BtuB crystal structure shows that upon TonB binding, R14 is removed from the protein interior so that the ionic interaction to D316 is broken. There is nothing in the structure to indicate that the R14A mutation alone would do anything unusual to BtuB that would not be accomplished by the binding of TonB. To address the concern about function of this mutant, we have performed a growth assay (Figure 3 —figure supplement 3) which shows that the BtuB R14A mutant functions in transport. This is also consistent with earlier work (Lathrop et al., (1995) J. Bacteriol. 177, 6810-6819) showing that a dipeptide insertion that would break the D316-R14A ionic bond does not destroy transport. We have included this information at the Results section. Some of the earlier work from Kadner’s laboratory shows that BtuB is highly tolerant to substitution and insertion mutations.

Regarding the incorporation of the spin label. We would like to point out that two spin labels – at site 90 and 93 – show the same effect. There is the possibility that both are modifying the protein in the same way, but this seems unlikely. Second, spin labels buried into proteins can alter folding energies, but they are generally well-tolerated in proteins and in general they do not dramatically alter structures (Altenbach et al., (2015) Methods in Enzymology, 564, 60). We have published structures with the R1 side chain at position 10 in the protein interior of the protein (Freed et al., (2010), Biophys. J. 99, 1604-1610). Except for some differences in B-factors, the structure is similar to the wild-type structure. Finally, regarding the incorporation of the spin label at site 90, site 90 on BtuB has been derivatized with a wide range of side chains (Mills et al., 2016, BBA 1858, 3105-3112). While these modifications lower B12 affinity by increasing the off rate, the substrate is still able to bind.

We fully admit that the structural changes we observe may or may not be directly related to transport and that more work is needed. But there has been relatively little progress on understanding the molecular details of transport in this system and that the movements that are seen here in SB3 on the C-terminal side of the core suggest that alternate mechanisms should be examined (other than an unfolding of the N-terminus of the core).

The reviewer appears to be uncomfortable with the fact that we attribute changes in the native environment to LPS. We admit that there may be other sources of the differences between the native and reconstituted system, but we do not have a problem proposing that LPS may make the difference. First, BtuB is known to be a highly alloteric protein, with conformational shifts coupled through various regions of the core. Second the computation work of Balusek and Gumbart (Biophys. J. 111, 1409-1417), provides evidence for an energetic coupling of LPS-loop interactions, the ionic interactions between the core and the barrel, and the Ton box. These interactions are consistent – at least qualitatively – with the observations that we have made.

Lastly, like any effort where we see something new, there is always the possibility that we have incorrectly interpreted the experiment. And there is always more to do. But we do not understand why the reviewer insists on raising the possibility of a complicated solution when a relatively simple explanation is available.

3. The authors sometimes misinterpret, misquote or neglect the literature. For example, they ignore the work that involved Jim Feix's lab, on SDSL of FepA 25 years ago, as well as other data on FepA, that demonstrated conformational motion akin to what they now describe for BtuB, both in vitro (Liu et al., 1995) and in vivo (Jiang et al., 1997). Their statement "…Computational work suggests.… that environment, specifically LPS, may be important in controlling the configuration of SB3" neglects other work on TBDT that demonstrated their completely different properties in vitro and in vivo. FepA has >100-fold lower affinity for ferric enterobactin when purified and reconstituted into detergent or liposomes (Payne et al., 1997), than when measured in bacterial cells (Newton et al., 1999). It's more difficult to rationalize an effect of LPS on SB3, that is ensconced with the 22-stranded β-barrel of BtuB, and thereby isolated from interactions with LPS.

We are sorry that the reviewer feels this way, and perhaps we have neglected a study that should have been sited. We are aware of this earlier work, but we did not site this work here because we do not believe it is relevant to the present study. Indeed, we have left out a number of references to earlier work (including our own) because they are not particularly relevant to the present study.

The reviewer mentions Liu et al. (1994), which provides evidence for a change in protein conformation in the ligand binding domain. The exact nature or size of this change was not possible to determine from the CW spectra taken and no distance measurements were made. Moreover, the results were obtained prior to the crystal structure of FepA so that the exact placement of the labeled sites was not known. It is curious that the CW spectra are not consistent with what is expected from a loop region, and inspection of the crystal structure indicates that the sites examined are within the barrel or facing outward near the edge of the barrel at the membrane interface. While this was a groundbreaking paper at the time, we do not see how it is relevant to our current work.

The Jiang et al. 1997 paper describes the spin-labeling of a single site on FepA in vivo. This was an important study that demonstrated that ligand uptake could be followed in vivo by examining the EPR spectrum of a site sensitive to ligand binding. Significantly this was the first case of an in vivo EPR study on an outer membrane protein. However, the exact nature of the structural change observed was not known, and the label (as seen in the subsequent crystal structure) is directed outward from the protein barrel near the extracellular interface. in vivo, FepA can transport, while in vitro it cannot (TBDT has never been reconstituted). As a result, it is perhaps not surprising that an in vivo measurement shows evidence for transport whereas an in-vitro experiment does not. We did not think that referencing this work added to the understanding of our data since it does not reveal significant details of the transport mechanism.

The other two papers Payne, 1997 and Newton, 1999, address ligand affinity, the kinetics of FepA uptake, and the effect of loop deletions on binding and ligand uptake. In the Newton 1999 paper, the difference in ligand affinity that is observed between the in vitro and in vivo cases is attributed to ligand depletion, which presumably results from the different methods used to make the binding measurements. The difference may be due to the difference between the in vivo and in vitro environments, but this was not the reason emphasized in the paper.